# A Federated Learning Approach to Breast Cancer Prediction in a Collaborative Learning Framework

**DOI:** 10.3390/healthcare11243185

**Published:** 2023-12-17

**Authors:** Maram Fahaad Almufareh, Noshina Tariq, Mamoona Humayun, Bushra Almas

**Affiliations:** 1Department of Information Systems, College of Computer and Information Sciences, Jouf University, Al Jouf 72311, Saudi Arabia; mfalmufareh@ju.edu.sa; 2Department of Avionics Engineering, Air University, Islamabad 44000, Pakistan; noshina.tariq@mail.au.edu.pk; 3Institute of Information Technology, Quaid-i-Azam University, Islamabad 45320, Pakistan; balmas@qau.edu.pk

**Keywords:** federated learning, breast cancer, prediction, collaborative learning, deep neural networks

## Abstract

Breast cancer continues to pose a substantial worldwide public health concern, necessitating the use of sophisticated diagnostic methods to enable timely identification and management. The present research utilizes an iterative methodology for collaborative learning, using Deep Neural Networks (DNN) to construct a breast cancer detection model with a high level of accuracy. By leveraging Federated Learning (FL), this collaborative framework effectively utilizes the combined knowledge and data assets of several healthcare organizations while ensuring the protection of patient privacy and data security. The model described in this study showcases significant progress in the field of breast cancer diagnoses, with a maximum accuracy rate of 97.54%, precision of 96.5%, and recall of 98.0%, by using an optimum feature selection technique. Data augmentation approaches play a crucial role in decreasing loss and improving model performance. Significantly, the F1-Score, a comprehensive metric for evaluating performance, turns out to be 97%. This study signifies a notable advancement in the field of breast cancer screening, fostering hope for improved patient outcomes via increased accuracy and reliability. This study highlights the potential impact of collaborative learning, namely, in the field of FL, in transforming breast cancer detection. The incorporation of privacy considerations and the use of diverse data sources contribute to the advancement of early detection and the treatment of breast cancer, hence yielding significant benefits for patients on a global scale.

## 1. Introduction

Breast cancer has emerged as a dominant global health concern, surpassing even lung cancer in prevalence [1] and ranking second in terms of mortality, making it the second leading cause of death among women worldwide [2]. The World Health Organization (WHO) recognizes breast cancer as the most commonly diagnosed cancer globally, accounting for 12% of all new cancer cases [3,4]. This formidable health challenge primarily affects women between the ages of 35 and 45, characterized by complex biochemical nuances. Detecting invasive breast cancer at its early stages offers the prospect of cure, particularly for women over 40, yet early detection remains a significant challenge, especially when relying on mammography as a diagnostic tool [5,6]. Despite its ubiquity and cost-effectiveness, mammography often falls short of timely diagnoses. Successful breast cancer diagnosis with mammography frequently occurs at later stages, contributing significantly to high mortality rates [7]. Advances in pharmaceutical research have improved treatment outcomes and reduced adverse effects, turning breast cancer into a malignancy with a more favorable prognosis. Consequently, the approach to breast cancer has evolved into a chronic condition, giving rise to a comprehensive strategy encompassing screening, diagnosis, therapy, and post-diagnosis follow-up [8,9]. Therefore, the early detection of breast cancer plays a pivotal role in this comprehensive strategy, potentially reducing mortality rates and increasing the likelihood of successful treatment. Various imaging modalities, including mammography, ultrasonography, Magnetic Resonance Imaging (MRI), and Positron Emission Tomography (PET), have been integral to breast cancer diagnosis [10]. Figure 1 shows the average ratio of increasing breast cancer cases, conventional diagnosis ratio, computer-aided diagnosis ratio, and AI-based diagnosis for the years 2018 to date. The data are acquired from Google Trends https://trends.google.com/trends/ (accessed on 24 September 2023).

Nevertheless, the substantial volume of imaging data generated during this diagnostic process poses a significant burden on radiologists, and diagnostic precision can be hindered by suboptimal or ambiguous images [11]. In addition, computer-aided diagnosis (CAD) is valuable for swiftly detecting and assessing breast cancer cases. This comprehensive approach encompasses three key stages: the initial identification and localization of potential tumors or cancer cells from preprocessed mammography, the extraction of tumor attributes (morphology, dimensions, radiodensity, prospective mass, and consistency), and final categorization into benign or malignant [12]. Traditional X-ray and mammography have given way to more advanced detection systems, such as data mining, neural networks, and Artificial Intelligence (AI) [13]. It is the field of study that focuses on creating intelligent machines that can perform tasks that require human intelligence [14]. AI systems can recognize patterns and adapt to new situations by analyzing data. They can also solve problems, make decisions, perceive their surroundings, and interact with humans and other machines naturally [15]. A creative and powerful application of AI to healthcare is presented in [16]. Timely and accurate breast cancer detection minimizes unnecessary treatments and improves patient survival rates. Machine Learning (ML) techniques have shown promise in this context; however, there is often a gap between the medical knowledge of professionals and their understanding of ML methods and results [17,18]. ML offers resilience in addressing diagnostic limitations stemming from the subjectivity of human practitioners.

Moreover, ML may represent the most economically viable breast cancer screening alternative in resource-constrained developing nations with large populations and limited healthcare resources. Recent advances in Self Supervised Learning (SSL) show promise in performance by leveraging extensive unlabeled datasets, reducing the need for annotations. However, obtaining substantial and diverse medical data remains critical for effectively training robust medical ML algorithms [19,20]. Besides having huge and diverse datasets, processing capacity has emerged as a prominent limitation in training ML algorithms. The discipline of distributed systems has emerged due to the increasing demand for highly efficient computer resources, including processing power, memory, and storage space [21]. Within this specific field, distributed ML has surfaced as a framework in which algorithms are implemented and deployed across multiple nodes, leveraging a larger number of data and processing capabilities and improving performance and efficiency. The core tenet of distributed learning entails the dispersion of algorithms among computer nodes, as expounded in [21]. Nevertheless, it is crucial to acknowledge that these operations are carried out without considering any potential limitations that might have to be enforced by these nodes. For example, the data employed across those nodes could come from disparate distributions. Consequently, many practical applications in collaborative learning must adhere more to the fundamental principle of Independent and Identically Distributed (IID) data among nodes. This encompasses situations when client information from mobile devices or healthcare data from various geographic and demographic parameters are implicated.

Federated Learning (FL) is acknowledged to be a collaborative learning paradigm that adeptly addresses a range of practical difficulties and sets itself apart from traditional distributed learning situations. It is delivering efficient enhancements in the healthcare domain, showcasing significant improvements [22]. It is crucial to consider several factors to ensure the accuracy and reliability of data analysis in a distributed network. Firstly, the statistical heterogeneity of data across nodes needs to be considered. It means acknowledging and accommodating variations in data characteristics and distributions among nodes. Secondly, addressing data imbalance across nodes is essential. It involves mitigating the situation where certain nodes possess significantly more or less data than others. Strategies should be implemented to balance the data distribution across nodes to avoid biased results. Thirdly, the limited communication capabilities within the distributed network must be managed effectively. Challenges such as loss of synchronization and variability in communication capabilities must be addressed to ensure seamless and accurate data transmission and processing. Lastly, handling the potential scenario of many nodes relative to the available data is important. It requires careful consideration of the distributed system’s scalability, privacy, and efficiency, ensuring that the analysis can handle the increased complexity and volume of data generated by numerous nodes [23].

The following are the contributions of this paper:An FL-based framework is presented for the categorization and early detection of breast cancer using DNN to train, classify, and detect breast cancer collaboratively.Careful data augmentation and class balancing is done using the Synthetic Minority Over-sampling Technique (SMOTE) on a substantial and diverse dataset of patients with breast cancer for testing and validation.Benchmark results are presented using a comprehensive set of evaluation measures to demonstrate the performance and efficiency of the proposed model.

The ensuing sections of the article are structured as follows: the literature review is presented in Section 2. The methodology and its mathematical modeling, along with the dataset, are explained in Section 3. Section 4 discusses the experimental results, and, finally, the conclusion is provided in Section 6.

## 2. Related Work

The current body of literature encompasses a substantial range of studies about breast cancer that hold important relevance. Sakib et al. [24] did a comparative analysis to investigate the application of ML and Deep Learning (DL) approaches in the context of breast cancer detection and diagnosis. The task of classification involved the utilization of five well-established supervised ML methodologies, namely, the Support Vector Machine (SVM), K-Nearest Neighbour (KNN), Decision Tree (DT), Logistic Regression (LR), and Random Forest (RF), in conjunction with a DL methodology. The solutions were assessed by implementing tenfold cross-validation on a dataset owned by the organization, which comprised 699 samples. The investigation conducted by the researchers indicates that the artificial artificial neural network (ANN) exhibits the highest degree of accuracy, achieving a noteworthy 98.57%. In contrast, it is noteworthy that both the RF and LR models demonstrated a commendable accuracy rate of 95.7%. Singh et al. [7] conducted a comparison investigation to evaluate different machine-learning approaches in the context of breast cancer diagnosis. Furthermore, the research paper presents a novel auto-encoder model that performs unsupervised breast cancer identification. The aim is to determine a succinct portrayal of traits that have a notable association with breast cancer. The utilized approaches were implemented using the openly available Breast Cancer Wisconsin (Diagnostic) Dataset, accessed through the Kaggle platform. The auto-encoder performs better than its counterparts, with a precision and recall rate of 98.4%. The assessment was conducted using various performance indicators, including accuracy in classification, recall, specificity, precision, False-Negative Rate (FNR), False-Positive Rate (FPR), F1-Score, and Matthews Correlation Coefficient (MCC). Based on the experimental findings, it can be shown that, after making the required modifications, the RF framework demonstrated superior performance compared to all alternative models. The model attained a rate of accuracy of 96.66% and an F1-Score of 96.30%.

FL has seen significant expansion since its establishment in 2016. This expansion encompasses collaborative learning and knowledge fusion scenarios involving several organizations [25]. FL has three major subcategories: Federated Transfer Learning, Horizontal FL, and Vertical FL. All three subtypes conform to the essential concepts of FL, which entail decentralizing data pooling through weight sharing and aggregation among multiple clients and a global server. These entities can be distinguished by the discrepancies discovered in their data sources. FL has been identified as a secure and efficient approach for addressing data isolation and labeling challenges, as highlighted in [26,27]. This technique enables the development of ML models by enabling them to integrate and consolidate information from multiple entities while ensuring that the data remains limited to its sources. The researchers employed a client–server architecture in their FL strategy, integrating Federated Averaging (FedAvg) as suggested by [23]. The FedAvg algorithm integrates decentralized Stochastic Gradient Descent (SGD) on local nodes with a centralized server that conducts model averaging. However, the input mammograms were drastically downsampled in the study conducted in [28]. While low resolutions may be deemed satisfactory for density classification, the decrease in detail has a detrimental impact on the accurate classification of malignancy. Additionally, no domain adaptation strategies were employed in this investigation to address the domain shift arising from variations in pixel intensity distributions. In the study by Jimenez et al. [29], a distinct methodology was employed, wherein high-resolution mammograms were utilized in conjunction with federated domain adversarial learning. Furthermore, the researchers implemented curriculum learning in the context of FL to enhance the performance of classification tasks. The proposed methodology enhanced the alignment across various domains and effectively resolved the challenge of domain shift by employing federated adversarial adaptation of domains. This work employs three different Full Field Digital Mammography (FFDM) datasets. It presents experimental results that support the effectiveness of the proposed memory-aware curricular method in improving classification performance.

Ma et al. [30] employed a hybrid approach, integrating the FL framework with Convolutional Neural Networks (CNN), in order to create a federated prediction model. This study showcased improvements in overall modeling and simulation conditions for five distinct forms of cancer. The cancer data exhibited a degree of accuracy that was above 90%, thereby outperforming single-model machines, tree models, linear models, and neural networks. Nevertheless, this work only compared several models beyond MLP. Additionally, it failed to address the concern of data imbalance and its corresponding treatment. Tan et al. [31] proposed developing an FL system that focuses on extracting features from individual participant settings instead of a Centralized Learning (CL) system. The research paper presents several innovative contributions. The initial stage employs transfer learning to retrieve features of data from the Region of Interest (RoI) from an image. This strategy aims to facilitate meticulous pre-processing and data augmentation for training. Additionally, the research in [31] employs the Synthetic Minority Oversampling Technique (SMOTE) for data processing. This technique aims to attain a greater level of consistency in data classification and improve the accuracy of diagnostic predictions for diseases. In addition, the research utilizes the FeAvg-CNN + MobileNet model within the FL framework to safeguard client privacy and personal protection. Finally, the work offers empirical findings derived from DL, transfer learning, and FL models. These models were evaluated using balanced and imbalanced datasets in the domain of mammography. Table 1 shows the comprehensive literature review summary.

The authors in [25] have designed an FL framework to classify breast cancer based on histopathology pictures. This study suggests using an FL methodology to tackle security issues effectively. The proposed concept entails the dissemination of patient model parameters in order to facilitate the integration of knowledge. Hence, the application of FL led to the exclusion of data sharing, and the inquiry was carried out employing the BreakHis dataset. The simulation study results indicate that the built FL model exhibited classification performance on par with centralized learning. Abunasser et al. [32] introduced a DL-based Xception model to enhance breast cancer detection and categorization/classification. The dataset was initially acquired using the Kaggle repository and afterward underwent pre-processing to eliminate unwanted noise. Subsequently, the dataset was partitioned into three distinct categories: validation, training, and testing.

The created exception model received training using the provided training dataset. In the training phase, a data augmentation method was employed to address the issue of overfitting. The layers of the exception model are tasked with extracting the most significant features, which are subsequently utilized by the ultimate softmax classification process to categorize breast cancer. The findings from the simulation analysis indicate that the created model exhibited a significant level of proficiency. Nevertheless, it is important to acknowledge that there was a rise in the duration of processing. Ogier et al. [33] focused on predicting the incidence of triple-negative breast cancer within the framework of FL. In order to guarantee the protection of the confidentiality of the data, this study implemented the concept of FL. The utilization of FL has effectively safeguarded the confidentiality and integrity of the patient’s data. The study proves that conventional ML techniques mostly utilized entire slide images to detect Neoadjuvant Chemotherapy (NACT) responses. Nevertheless, the collective training of ML techniques enhances the effectiveness in detecting triple cancer. Moreover, this study provides evidence to support the assertion that the FL paradigm is well-suited for implementation in real-time applications.

This literature review provides a comprehensive overview of multiple studies pertaining to the detection and diagnosis of breast cancer. It specifically emphasizes the utilization of machine learning and federated learning methodologies. These studies exhibit encouraging outcomes in relation to precision and efficacy. Nevertheless, it is common for these methodologies to have limited comparisons with alternative approaches, and they may not adequately consider issues related to scalability, data imbalance, and privacy. Although these methodologies demonstrate promise in augmenting the detection of breast cancer, additional study is needed to remove their inherent constraints and render them more applicable in real-world clinical settings.

## 3. Proposed Methodology

The dataset utilized in this study is the Breast Cancer Wisconsin (BCW) Diagnostic dataset with dataset details in Table 2. The dataset consists of 569 observations and 32 variables. For instance, the dataset has 569 patients and 32 columns. These columns include a patient ID, 30 attributes derived from 10 morphological qualities, and a final column denoting whether the patient has been diagnosed with malignant or benign tumors (see Table 3 for patient data details). The pathologists have conducted a preliminary analysis of these traits and then recorded them in tabular form. It is imperative to acknowledge the necessity of including the worst-case values for all ten cell nuclei morphological features in each patient [34]. Among the 569 cases, the class distribution is as follows: 357 cases belong to the Benign class, marked by the numerical value 0, whereas 212 cases belong to the Malignant class, denoted by the numerical value 1. Furthermore, it was determined that none of the columns exhibited any missing values. Therefore, it is necessary to consider the presence of 30 input features in order to gain a comprehensive understanding of constructing a classification prediction model. Furthermore, it may be argued that a well-balanced and non-skewed distribution of class labels contributes to the favorable convergence of loss in ML algorithms, ultimately leading to the attainment of appropriate solutions [35].

According to recent reports, evidence suggests that an increased number of independent features contribute to the development of ML models that exhibit a reasonable level of accuracy when fitting the dataset [36]. Nevertheless, the inclusion of a substantial number of characteristics, particularly in cases with inadequate data points, hinders the generalization capability of prediction models [36]. Additionally, multi-collinearity may result in confounding effects and contribute to increased uncertainty when predicting the parameter values of the FL model. The BCW Diagnostic dataset exhibits a notable disparity in the number of features, namely, 30, compared to the diagnosis outcomes of 569 individuals, including benign and malignant classifications. Table 4 represents the complete set of 30 features of the BCW diagnosis dataset along with their description.Hence, a two-step method should be employed, involving feature selection techniques in conjunction with the ML classification model. The conventional approach involves utilizing all available features for training the classification model. In contrast, our proposed framework employs a two-step process. In the first stage, 30 features undergo a feature selection approach named L1 Regularization (L1) to identify the most relevant features. These optimal features are then utilized for training the classification model. Subsequently, the ML model, namely Deep Neural Network (DNN), is trained and assessed using the feature sets obtained from the feature selection technique. The selection procedure is carried out by utilizing essential classification assessment criteria, such as Accuracy, overall Precision, and Recall (particularly for malignant cases), on a distinct set of test data.

### 3.1. Data Pre-Processing

In this phase, we acquire and preprocess the data for FL, as shown in Equations (Equation 1)–(Equation 3). Each client *i* has its local dataset, denoted as $D_{i}$, where $D_{i}=\left(X_{i},y_{i}\right)$. Here, $X_{i}$ represents the feature matrix containing input data and $y_{i}$ is the label vector containing target values. Data cleaning is an essential step, which involves addressing missing values ($M_{missing}$), handling outliers ($O_{outliers}$), and identifying and eliminating duplicates. The resulting cleaned dataset is denoted as $D_{cleaned}$. Additionally, label encoding is applied to convert categorical labels into a numeric format. We use a label encoder *L* to perform this transformation, producing encoded labels ${\hat{y}}_{i}$ from the original labels $y_{i}$.
(1)$$D_{i}=\left(X_{i},y_{i}\right)\;$$
(2)$$D_{cleaned}=Clean\left(D_{i},M_{missing},O_{outliers}\right)\;$$
(3)$${\hat{y}}_{i}=L\left(y_{i}\right)$$

In this study, a comprehensive integration of the following preprocessing procedures has been undertaken in order to enhance the data preparation process. The dataset preprocessing steps involve a systematic approach to refine and enhance the quality of the BCW (Diagnostic) dataset before its utilization. Each step serves a specific purpose in addressing challenges such as outliers, class imbalance, and limited data diversity. Table 5 summarizes the steps and values involved in data preprocessing.

Original Dataset Distribution: The initial dataset consists of 569 instances, with 357 instances categorized as Benign (B) and 212 as Malignant (M). This distribution forms the foundation for subsequent preprocessing steps.Removal of Outlier: Identifying and eliminating outliers is a critical step in the preprocessing stage, wherein outliers are methodically identified and addressed based on the statistical characteristics of the dataset. The dataset has been analyzed to determine important statistical measures, including the first quartile (Q1), the third quartile (Q3), and the interquartile range (IQR). The IQR is a statistical metric that offers significant insights into the dispersion of a given dataset. The calculation involves determining the disparity between the Q1 and the Q3. Q1 represents the 25th percentile, and Q3 represents the 75th percentile. The IQR is calculated as Q3–Q1 with a threshold for identifying potential outliers 1.5 × IQR. The data points are considered potential outliers if they fall below the Q1 − Threshold or above the Q3 + Threshold. Table 6 represents the IQR for each parameter. One pivotal element of this methodology entails the establishment of a threshold, which has been delineated as the equivalent of 1.5 times IQR, functioning as a criterion for detecting probable outliers. In statistical analysis, outliers are commonly defined as data points that lie beyond the range of (Q1 − 1.5 times the interquartile range) or (Q3 + 1.5 times the interquartile range). Following the conclusion of the identification phase, the dataset has been effectively purged of any outliers. After removing outliers, the number of observations is 538.After Outliers Removal: To ensure that extreme values do not unduly influence the model, outliers are removed from the dataset. This refinement results in a reduced but more robust dataset comprising 538 instances, with 340 instances labeled as Benign (B) and 198 as Malignant (M).Data Augmentation: Data augmentation has been employed to expand the dataset by generating extra data samples, hence obviating the need to gather fresh real-world data. Gathering large-scale annotated datasets proves to be exceedingly challenging given the scale that is required. Augmentation stands out as a highly effective interface for impacting the training of Neural Networks [37]. Table 7 and Table 8 show the dataset before and after augmentation values. For instance, the values of Min and Max for ‘smoothness_se’ in Table 7 are 0.0 and 0.03, respectively, whereas, in Table 8, the values for Min and Max are −0.01 and 0.03, which represents instances with smoother textures. This augmentation introduces a new dimension to the feature, allowing the model to recognize cases with even lower smoothness. Similarly for ‘concavity_worst’, Table 7 shows a Min value of 0.0 and a Max of 0.77, whereas Table 8 shows a Min of −0.01 and a Max of 0.78. This indicates a subtle change in the representation of worst-case concavity. This augmentation introduces a nuanced variation in the dataset, potentially improving the model’s ability to generalize to a broader range of scenarios. These differences in minimum and maximum values showcase the impact of data augmentation in capturing more diverse and nuanced patterns within the dataset, which can contribute to improved model robustness. Likewise, several characteristics display changes in their statistical qualities, indicating the impact of the enhancements. Specifically, attributes such as mean, standard deviation, minimum, and maximum values exhibit variations that signify the augmentation effects. The augmentation process entails the deliberate introduction of controlled variations or perturbations to the existing data points, expanding the dataset’s size and diversity. Implementing the data augmentation method involves the introduction of controlled random noise to the feature values of each data point, enhancing its practicality. To maintain the inherent characteristics of data, it is conventional to ensure that the amplitude of the noise remains minimal.

The enhanced data samples are subsequently produced by introducing alterations to the original data points using this noise while ensuring that their target labels remain unchanged. By incorporating these variances into the dataset, the methodology improves the model’s capacity to acquire knowledge from the data and apply it to samples that have yet to be previously encountered. This procedure helps to reduce the likelihood of overfitting and promotes the enhanced performance of the model, especially in scenarios where the initial dataset may have limitations or imbalances. Data augmentation techniques are employed to enrich the dataset and expose the model to a broader range of scenarios. To expand the dataset’s size, we introduced noise ranging from 0 to 0.01 into the data. This resulted in a total of 5742 records. The augmented data contained 3212 Benign and 2530 Malignant records.

Adding Noise: Noise, ranging from 0 to 0.01, is introduced into the dataset. This augmentation significantly increases the dataset size to 5742 instances, with 3212 instances representing Benign (B) and 2530 instances representing Malignant (M). The addition of noise contributes to the model’s ability to generalize by exposing it to a more diverse set of data points [38].SMOTE for Class Balance: Synthetic Minority Over-sampling Technique (SMOTE) is applied to address class imbalance [39]. This technique involves generating synthetic samples for the minority class to balance class distributions. After applying SMOTE, the dataset achieves balance with 3212 instances for both Benign (B) and Malignant (M) classes, resulting in a total of 6424 instances. This balance is crucial for training a machine learning model that is not biased toward the majority class.

The dataset preprocessing steps collectively contribute to creating a more reliable, diverse, and balanced dataset. The refined dataset, encompassing various instances of Benign and Malignant cases, is poised for effective utilization in subsequent machine learning models, particularly deep neural networks, where the quality of the input data is paramount to the model’s performance and generalization capabilities.

### 3.2. Feature Selection

It reduces the number of input features by eliminating those that are not significant. Selecting the appropriate features helps reduce the computational cost, improves model performance, and reduces overfitting [40]. The selection of pertinent characteristics holds significant significance in enhancing the precision and comprehensibility of ML models, specifically in binary classification for breast cancer diagnosis. The primary aim of this phase was to distinguish between cancerous (1) and benign (0) conditions within the Breast Cancer Dataset. It was achieved by carefully choosing and defining a subset of qualities that provided the most relevant and useful data. By carefully choosing a suitable combination of features, we have effectively enhanced the model’s capacity to classify breast cancers while preserving its comprehensibility and simplicity. The primary aim of the feature selection procedure utilized in this study’s methodology was to determine the most relevant features in the binary classification task of cancer identification. The first version of the Dataset included 30 features, each containing unique information related to different aspects of tumor characteristics.

To determine the optimal set of features, we employed a feature selection approach that integrates L1 (Lasso) regularization into a DNN framework. L1 regularization allocates coefficients to individual features, thereby quantifying their impact on the predictive outcomes of the model. Features that had non-zero coefficients were deemed crucial for the classification job. In contrast, features with coefficients lowered to zero were seen as less significant and eliminated from the model. The rigorous feature selection method yielded a total of 25 features that were deemed essential for the proper classification of breast cancers. The parameters mentioned above encompass crucial tumor attributes that substantially differentiate between malignant and benign cases. The 25 features were chosen based on their ability to enhance the predicted accuracy of the model while minimizing its dimensions and complexity. There are numerous advantages associated with feature selection in the given situation. Firstly, implementing this approach ensures that the model prioritizes the most pertinent qualities, enhancing its ability to differentiate between different entities or categories.

Additionally, it aids in mitigating overfitting by eliminating extraneous or repetitive features that impede the model’s ability to generalize to unfamiliar data. Furthermore, the utilization of a limited number of variables contributes to the interpretability of the model, enabling healthcare practitioners to discern the tumor characteristics that hold the greatest significance in identifying the presence of cancer. Here is the mathematical modeling of the feature selection methodology:

#### 3.2.1. Feature Selection Using L1 Regularization for Breast Cancer Classification

L1 (Lasso) regularization in DNN is used to perform feature selection for breast cancer classification based on Equation (Equation 4).
(4)$$\mathrm{Minimize}\;J\left(\theta \right)=-\frac{1}{m}\sum_{i=1}^{m}\left(y^{\left(i\right)}\log\left(h(\theta^{T}x^{\left(i\right)})\right)+\left(1-y^{\left(i\right)}\right)\log\left(1-h(\theta^{T}x^{\left(i\right)})\right)\right)+\lambda {∥\theta ∥}_{1}$$
where *m* is the number of data samples; $x^{\left(i\right)}$ represents the feature vector for the *i*th sample; $y^{\left(i\right)}$ is the binary targeted variable 1, which signals cancer, and 0, which does not. In addition, θ reflects LR model coefficients weights for each feature, $h(\theta^{T}x^{\left(i\right)})$ is the sigmoid function maps linear feature combinations $\theta^{T}x^{\left(i\right)}$ to a probability between 0 and 1, λ reflects the L1 regularization strength parameter, and ${\left|\theta \right|}_{1}$ is the L1 norm (sum of absolute values) of the weight vector θ. Reducing the aforementioned objective function promotes the tendency for numerous feature weights to attain a value of zero, efficiently choosing a subset of characteristics that significantly contribute to the classification task. The features that have been chosen offer significant insights into the diagnosis of breast cancer.

#### 3.2.2. Train, Test, and Validate Dataset

In order to implement FL, the global dataset is carefully divided into distinct subsets that are specific to each client. Every individual client is provided with a specific dataset, guaranteeing the utmost protection of data privacy. A 70:15:15 partitioning strategy is utilized to divide the data into training, testing, and validation sets, facilitating localized model training under our federated system. A randomized order of data instances is shuffled and disseminated to all participating clients.

### 3.3. Model-Building

Using DNNs, client-side training enables each client to train localized models on their datasets. This decentralized method adds to the global federated model’s collective intelligence in addition to protecting data privacy.

Deep Neural Network Framework: In this phase, each client *i* trains its local model, denoted as $M_{i}$, using a DNN. The DNN takes the feature matrix $X_{i}$ and the encoded labels ${\hat{y}}_{i}$ as input. This local model $M_{i}$ is trained independently on the client’s data, allowing each client to capture local patterns and knowledge, as shown in Equation (Equation 5).
(5)$$M_{i}=\mathrm{DNN}\left(X_{i},{\hat{y}}_{i}\right)\;$$The primary element of the client-side training process is the meticulously designed DNN, specifically tailored to address the binary classification challenge. The architectural architecture illustrated in Figure 2 consists of a solitary input layer and three hidden layers, along with one output layer, with each holding unique and well-defined attributes.Input Layer: The input layer is designed to align with the dimensions of the feature vectors present in the dataset, successfully adapting the unique qualities of the data.Hidden Layers: The DNN is composed of three hidden layers that have been specifically developed to methodically extract and represent increasingly intricate features from the input data. Each hidden layer has a form of connectivity known as dense connectivity. We used Rectified Linear Units (ReLU) as the activation function. It enhances the level of non-linearity, hence facilitating the model in efficiently capturing complex relationships that exist within the data.Fully Connected Layer: This layer, sometimes referred to as a dense layer, serves as an intermediate stage connecting the hidden layers as well as the output layer. The fully connected layer, similar to the hidden layers, employs the ReLU function for activation. The fully connected layer has a carefully selected number of neurons that enable efficient feature transformation while ensuring the model remains manageable.Output Layer: The final layer of our DNN-based FL design, known as the output layer, is of utmost importance in producing predictions for binary classification. This layer comes with the Sigmoid activation function, which is frequently employed in binary classification applications. The application of this activation function leads to the production of probabilistic outputs that cover the numerical interval from 0 to 1, hence facilitating the estimation of class probabilities. The output layer comprises two neurons, each representing one of the two binary classes. The utilization of the Sigmoid activation function enables individual neurons to evaluate the likelihood of membership within their corresponding class autonomously.

### 3.4. Model Compilation

The model compilation process encompasses the meticulous specification of the optimizer and loss function.

Selection of Optimizer: We utilize the Adam optimizer due to its effectiveness in gradient-based optimization [41]. The learning rates are dynamically adjusted in order to accelerate the process of convergence.Loss Function: As seen in our code implementation, the sparsely categorical cross-entropy loss function was selected. This option is suitable for problems involving binary classification since it measures the discrepancy between the expected and actual class labels [42].

### 3.5. Training Parameters

The selection of training parameters plays a crucial role in attaining both model convergence and generalization. The parameters are carefully configured in order to examine their influence on the training of the model. A variety of epochs (i.e., 5–70) are utilized to investigate different training scenarios.

Number of Epochs: We systematically investigate the impact of varying the number of training epochs. This range of exploration enables us to examine the behavior of the model across different stages of training, encompassing both rapid convergence and prolonged periods of refinement.Batch Size: The code implementation uses a batch size of 32. The selection of this particular configuration is made with the intention of achieving a harmonious equilibrium between computing efficiency and the stability of the model. The modification of batch size can significantly affect the rate at which convergence occurs and the amount of memory used, making it a crucial factor that requires careful deliberation in practical applications.Early Stopping Mechanism: The implementation of an early stopping mechanism, with a patience of 10, is crucial in mitigating the issue of overfitting. The patience parameter denotes the duration, measured in epochs, during which training will continue without any improvement in validation loss before it is terminated. The value is chosen carefully to guarantee that the model terminates training when additional iterations are unlikely to result in substantial enhancements.

### 3.6. Model Training and Saving

The key aspect of the client-side training function lies in the utilization of the backpropagation technique for the iterative optimization of the DNN model. Throughout various periods, the model acquires the ability to establish a correlation between input characteristics and desired outcomes, gradually improving its capacity to make accurate predictions. Simultaneously, the training process diligently monitors the validation loss in order to identify any indications of overfitting and implements early stopping if necessary. After the completion of successful training, the trained DNN model is safely stored in an archive. The saved model files play a crucial role in the succeeding stages of our FL process. Significantly, every client produces and maintains its unique model file, which secures individual data privacy and autonomy while also contributing to the collective intelligence encapsulated inside the federated model. Nomenclature Section presents a quick description of the symbols used in the section.

Algorithm 1 represents the Local Model Training of the ML model $M_{i}$ at a given client. The input of the system is the local dataset $D_{i}=\left(X_{i},{\hat{y}}_{i}\right)$, where $X_{i}$ represents the feature matrices and ${\hat{y}}_{i}$ represents the encoded labels. The method progresses by undergoing a sequence of iterations, with each iteration comprising numerous epochs. To ensure unpredictability, the dataset is randomized inside each epoch. The technique of batching is utilized to partition the dataset into smaller segments. Within each batch, the local model calculates predictions ${\hat{y}}_{pred}$ using the present model weights. The loss function $L_{batch}$ is then computed to evaluate the model’s performance on the given batch. Following this, the weights of the model are adjusted by a weight update function, typically employing gradient descent, in order to minimize the loss. The procedure mentioned above is iterated for the designated number of epochs, yielding a trained local model denoted as $M_{i}$. The model, as mentioned earlier, acquires and incorporates client-specific patterns and information at the local level from its localized dataset, which will subsequently be employed in the context of FL.
**Algorithm 1** Local Model Training Algorithm  1: **Input:** Local dataset $D_{i}=\left(X_{i},{\hat{y}}_{i}\right)$, Neural Network architecture, number of epochs, batch size  2: **Output:** Trained local model $M_{i}$  3: **procedure** Local Model Training($D_{i}$, Neural Network, epochs, batch size)  4:    Initialize local model: $M_{i}$  5:    **for** each epoch in 1 to epochs **do**  6:        Shuffle $D_{i}$ randomly  7:        **for** each batch in $D_{i}$ with batch size batch_size **do**  8:            Extract batch data: $X_{batch}$, ${\hat{y}}_{batch}$  9:            Compute predictions: ${\hat{y}}_{pred}=M_{i}\left(X_{batch}\right)$10:           Calculate loss: $L_{batch}=Loss\left({\hat{y}}_{batch},{\hat{y}}_{pred}\right)$11:           Update model weights: $M_{i}.update\_weights\left(L_{batch}\right)$12:        **end for**13:    **end for**14: **end procedure**15: **Return** Trained local model $M_{i}$

### 3.7. Loop across Clients

Within the framework of our FL orchestration, each client autonomously performs the client training function—the procedure mentioned above results in the production of unique model files. The model files serve as repositories for the knowledge and insights acquired by individual clients throughout the training process. The division of these models highlights our firm dedication to protecting the confidentiality and accuracy of client data while also contributing to the overall enhancement of intelligence in our federated model.

### 3.8. Server-Side Model

Model Aggregation Function: The FL system we have developed concludes with the server-side model aggregation function FedAvg algorithm, which plays a crucial role in combining the varied insights obtained from individual clients [43]. In the implemented code, this particular function has been methodically designed and developed to accomplish its intended purpose effectively. The function for model aggregation on the server side is designed to accept client models as input and provide a federated model as output. The code is designed in a manner that explicitly facilitates the processing of models from different clients. An essential element of this function is its ability to produce federated forecasts. The procedure, as mentioned above, is accomplished by successively executing the client models, producing predictions for a cohesive set of inputs, and subsequently merging these forecasts. The strategy described above forms the basis of the collaborative process of decision-making within our federated architecture.To create a global model, server-side model aggregation is performed. This function aggregates the local models from all clients into a federated model, denoted as $M_{fed}$, as shown in Equation (Equation 6). The federated model is a collective representation of knowledge learned from all clients. It leverages the insights captured by individual clients during local training to make global predictions.
(6)$$M_{fed}=\mathrm{Aggregate}\_\mathrm{Models}\left(\left\{M_{1},M_{2},\ldots ,M_{n}\right\}\right)\;$$Federated Model: After model aggregation, the federated model $M_{fed}$ is ready for global training, see Equation (Equation 7). It is compiled with an optimizer, typically Adam, and a loss function, such as Sparse Categorical Cross-Entropy. The global training phase fine-tunes the federated model using the aggregated knowledge from all clients, improving its overall performance.
(7)$$M_{fed}.compile\left(optimizer\right)=Adam\left(\right),loss=Sparse\_Categorical\_CrossEntropy\;$$The fundamental element in our FL framework is around the server-side aggregation of client models. The procedure above entails the amalgamation of information derived from distinct client models in order to construct a unified federated model. The aggregation function utilized on the server side performs an iterative procedure that traverses the models provided by several clients, thereby combining their respective contributions. The aggregation approach is the process of adding together the predictions made by separate client models. It allows the federated model to utilize the collective knowledge obtained from diverse data sources. The federated model is built upon the aggregated forecasts, which are derived from the design of each of the local models. The federated model represents a unified embodiment of the combined intellectual capacity originating from multiple participating clients.Algorithm 2 illustrates the process of global model training involving the development and optimization on a global scale. The method of global model training involves the aggregation of predictive capabilities exhibited by individual client models within the framework of FL. The process commences by collecting forecasts from individual client models using validation data. The predictions mentioned above are aggregated to provide a unified collection of predictions referred to as “client_predictions”. The outcome, denoted as global_prediction, is ascertained by selecting the class that possesses the highest probability. The resultant global federated model, referred to as Federated_Model, encompasses the collective knowledge derived from all clients involved in the process. The utilization of a collaborative approach significantly improves the model’s capacity to generalize to novel data and generate precise predictions while operating in a decentralized and privacy-preserving manner. Consequently, this renders it an influential tool for jobs involving FL.Final ClassifierThe final classifier used for making predictions is the federated model $M_{fed}$, shown in Equation (Equation 8). This model embodies the collective intelligence learned from all clients during the FL process. It serves as the ultimate decision-maker for tasks like classification and inference.
(8)$$Final\_Classifier=M_{fed}\;$$In the proposed Algorithm 3, the participation of each client in model training is based on its dataset $D_{i}$, which may be represented as $D_{i}=\left(X_{i},y_{i}\right)$. This dataset comprises feature matrices $X_{i}$ and label vectors $y_{i}$. Data preprocessing involves several important stages to assure the quality of the data, including the treatment of missing values ($M_{missing}$) and outliers ($O_{outliers}$). The process of encoding labels involves the utilization of a label encoder denoted as *L*, which subsequently produces encoded labels denoted as ${\hat{y}}_{i}$. The local client models, denoted as $M_{i}$, are trained in a separate manner using DNN with the input data $X_{i}$ and the corresponding predicted output ${\hat{y}}_{i}$. The variable *i* is in the range from 1 to *n*, representing each participating client and iterating over all clients in the federated learning setting. The aforementioned local models can collect patterns that are distinct to individual clients. The process of model aggregation involves the integration of individual local models into a federated model, denoted as $M_{fed}$, which serves as a representation of the collective knowledge. The global training of $M_{fed}$ is conducted, and it afterward functions as the ultimate classifier. The key symbols utilized in this study encompass $D_{i}$, $M_{missing}$, $O_{outliers}$, *L*, ${\hat{y}}_{i}$, $M_{i}$, and $M_{fed}$.

**Algorithm 2** Global Model Training
  1: **Input:** List of client models $\left[{Client}_{1},{Client}_{2},\ldots ,{Client}_{N}\right]$  2: **Input:** Number of clients *N*  3: **Output:** Global federated model Federated_Model  4: Initialize an empty list predictions  5: **for** i←1 to *N* **do**  6:    ${predictions}_{i}\leftarrow {Client}_{i}.predict\left(X\right)$           ▹ Make predictions on validation data  7:    $predictions.append({predictions}_{i})$                 ▹ Aggregate predictions  8: 
**end for**
  9: combined_predictions←sum(predictions)             ▹ Sum predictions across clients10: final_predictions←argmax(combined_predictions) ▹ Select class with the highest probability11: Initialize the global federated model FederatedModel12: 
**Return**

 FederatedModel




**Algorithm 3** Federated Learning Algorithm
  1: **Input:** Local datasets $D_{i}=\left(X_{i},y_{i}\right)$ for clients *i*, $M_{missing}$, $O_{outliers}$, label encoder *L*  2: **Output:** Federated model $M_{fed}$  3: **procedure** 
Data Acquisition and Pre-processing  4:    **for** each client *i* **do**  5:        Load local dataset $D_{i}=\left(X_{i},y_{i}\right)$  6:        Clean $D_{i}$ using $M_{missing}$, $O_{outliers}$  7:        Encode labels: ${\hat{y}}_{i}=L\left(y_{i}\right)$  8:    **end for**  9: 
**end procedure**
10: **procedure** 
Local Model Training11:    **for** each client *i* **do**12:        Initialize local model: $M_{i}$13:        Train $M_{i}$ using DNN with $X_{i}$ and ${\hat{y}}_{i}$14:    **end for**15: 
**end procedure**
16: **procedure** 
Model Aggregation17:    Initialize federated model: $M_{fed}$18:    Aggregate local models: $M_{fed}=\mathrm{Aggregate}\_\mathrm{Models}\left(\left\{M_{1},M_{2},\ldots ,M_{n}\right\}\right)$19: 
**end procedure**
20: **procedure** 
Global Model Training21:    Compile $M_{fed}$ with optimizer (e.g., Adam) and loss (e.g., Sparse Categorical Cross-Entropy)22:    Train $M_{fed}$ using global data23: 
**end procedure**
24: **procedure** 
Final Classifier25:    **Output:** Final classifier $M_{fed}$26: 
**end procedure**
27: 
**Return**

$\;M_{fed}$




### 3.9. Federated Learning Rounds

The FL technique, as seen in Figure 3, is structured into multiple rounds, where each round corresponds to a distinct iteration in the collaborative learning procedure.

The execution of FL rounds occurs iteratively. In the code implementation, the number of rounds is denoted as ‘NUM–ROUNDS’. Every iteration encompasses a sequence of client training and server-side aggregation. During each iteration of FL, each client autonomously trains their models using their respective local datasets. The training approach employed in this study utilizes the client training function, as outlined in the established methodology. After the completion of client training, the server-side model aggregation function is called. The preceding section elucidates a procedure that entails the synchronization of client models as well as the development of an enhanced federated model.

### 3.10. Evaluation Parameters

Accuracy: The performance of the classifier is evaluated using accuracy as a metric. The metric under consideration is the proportion of accurately identified observations in relation to the total number of observations. The value of the variable falls inside the interval of 0 to 1. In the context of measurement or evaluation, a precision value of 1 denotes a high or optimal level of accuracy. In contrast, a number of 0 represents the lowest or poorest level of accuracy. The mathematical computation of a model’s correctness is possible [44]:
(9)$$A_{CC}=\frac{\left(TR_{ps}+TR_{ng}\right)}{\left(TR_{ps}+TR_{ps}+FL_{ps}+FL_{ng}\right)}\;$$The variable $A_{CC}$ denotes the accuracy, whereas $TR_{ps}$ indicates the True Positive value in the label class, indicating instances where the model correctly recognizes a true label as true. The variable $TR_{ng}$ denotes the True Negative metric in the label class, indicating instances where the model correctly identifies incorrect labels. The variable $FL_{ps}$ denotes the False Positive metric, indicating instances where the model wrongly classifies a sample as false, whereas $FL_{ng}$ represents the False Negative metric, indicating instances where the model incorrectly classifies a sample as true. Table 9 represents the description of these variables in our case. It is important to aim for a balance between them as focusing too much on any one metric could lead to an imbalance in the others. For instance, trying to minimize $FL_{ng}$ (missing a cancer diagnosis) might increase $FL_{ps}$ (overdiagnosing cancer), which can lead to unnecessary treatments and stress for patients.Precision: Precision is defined as the ratio of the total number of correct positive predictive values to the total number of positive predictive values [45]. Additionally, it provides a value that is between 0 and 1. A precision of 1 indicates a highly predicted value or the best value, while a precision of 0 indicates the least predictive or the worst value. Accuracy can be measured using mathematical formulas given in Equation (Equation 10).
(10)$$\mathrm{pr}=\frac{TR_{ps}}{\left(TR_{ps}+FL_{ps}\right)}\;$$Recall: In addition, the recall bears the label Sensitivity. The accuracy rate is defined as the ratio of correctly detected positive values to the total number of samples that are classified as positive. The recall value is somewhere in the range of 0 to 1, where 1 indicates a strong recall and 0 indicates a lesser recall. The mathematical expression for it is shown in Equation (Equation 11).
(11)$$\mathrm{rc}=\frac{TR_{ps}}{\left(TR_{ps}+FL_{ng}\right)}\;$$F1-Score: The F1-Score accurately recalls the weighted average (also known as F1-Score). This score includes false positives and false negatives. Assigning more weight to precision and less to recall yields 0.75 beta. When recall is weighted greater than precision, the beta value is 2. We choose ‘1’ for beta to balance precision and recall. The F1-Score frequently outperforms accuracy, especially in unequal class distributions. Accuracy is best when false negatives and positives have similar costs. If false positives and false negatives have different costs, Precision and Recall should be considered jointly. Between 0 (lowest) and 1 (best), small precision or recall yields a low value. The harmonic mean of precision and recall determines it. Equation (Equation 12) represents its mathematical form.
(12)$$F1-Score=2\times \frac{\left(\mathrm{pr}\times \mathrm{rc}\right)}{\left(\mathrm{pr}+\mathrm{rc}\right)}\;$$

## 4. Experimentation Setup and Results

The proposed work is implemented on a system described as a PC with an Intel (R) Core (TM) GPU @3.20 GHz CPU, 8.0 GB RAM, and a Windows 10 operating system. Python is utilized as a programming language in these experiments, and Google Colab is used as an IDE. It is a web-based programming environment with much interactivity. Python is a simple, open-source, and effective programming language with plenty of open-source modules and community support to construct graphs and statistical models. We picked it because it is the most recent and reliable version of Python. A local host is created by Google Colab so that the code may execute on the browser. The details of a dataset and implementation of the proposed framework are explained in the section below.

The dataset undergoes a shuffling process, ensuring a randomized order of data instances. Subsequently, this uniformly shuffled dataset is disseminated to all participating clients. This meticulous approach is essential to maintain consistency and fairness in the initial distribution of data across the federated network. There were two local clients and one global model. Each model is evaluated on specific performance measures. Each model’s performance is evaluated using Accuracy, F1-Score, Precision, Specificity, and Recall. It describes the performance of classifiers on a set for which true values are known. It helps understand errors the classifier makes [46].

### 4.1. First Iteration Results

The results after the first iteration of our local and global model are given below.

#### 4.1.1. Local Model

Accuracy: During the preliminary phase, we conducted training and validation processes on the common augmented dataset, resulting in the development of two local models. The examination of the outcomes of each model is described below.The first performance of the first model depicted in Figure 4a exhibits encouraging indications. In the initial stages of training, the model exhibits a swift learning process and demonstrates adaptability to the provided training data, leading to a notable improvement in training accuracy. As anticipated, the model’s progress reaches a point of stabilization, suggesting that it is unable to enhance its performance on the training set further. The validation accuracy has a strong correlation with the training accuracy pattern but is continually trailing behind due to its exclusion from the model’s training procedure. At around 50s (continued to 70), both the training and validation accuracies exhibit stabilization, with the training accuracy converging to approximately 95% and the validation accuracy hovering around 90%. The observed convergence in this preliminary iteration indicates that the initial model successfully mitigates the issue of overfitting on the training data and exhibits a commendable capacity to generalize effectively to the novel, unobserved data. The results given in this study provide a strong foundation for the possible effectiveness of the model as a dependable predictor.Similarly, the second model depicted in Figure 4b demonstrates the same beginning trajectory. During the initials, the model demonstrates its ability to acquire knowledge from the training data efficiently, hence displaying its aptitude for capturing the fundamental patterns present within the dataset. As the training advances, it can be observed that both the training and validation accuracies exhibit a consistent pattern of convergence, which is consistent with the behavior observed in the initial model. At around 50 (continued to 70), the training accuracy achieves a level of approximately 95%, whereas the validation accuracy stabilizes at around 90%. It is worth mentioning that there is a short decline in the accuracy of validation at around the 10-epoch milestone. The initial iteration of the training process is attributed to the model’s dynamic learning and adaption. However, despite this initial performance setback, the second model, similar to the first one, exhibits the capability to efficiently generalize to unfamiliar data, indicating a promising start in its advancement.The initial iteration of our models demonstrates promising beginnings as they exhibit respectable performance, particularly given the constrained amount of training repetitions. Additional refinement and investigation will yield a more extensive comprehension of their capacities, yet these first findings indicate a promising outlook for these models as dependable prognosticators.Loss: In the initial iteration, we examined the performance of both ML models. The models utilized several training methodologies, with a specific emphasis on the implementation of data augmentation strategies. The initial model, as depicted in Figure 5a, shows loss analysis focused on the surveillance of misclassified cancer cases during the training process. The graph illustrates a persistent decline in loss over time, as observed in both the training and validation datasets. The current trend suggests that the model’s capacity to identify the cancer dataset for detection accurately is continuously improving. Significantly, the validation data demonstrated a marginally greater loss in comparison to the training data, as expected, due to the fact that the validation data were not encountered during the training process. The observed discrepancy in loss values serves as an indication of the model’s practical efficacy when applied to novel data, confirming that it did not excessively adapt to the training data.Furthermore, the observed decrease in loss on the validation data over time suggests that there is potential for additional improvement in the model. The marginal disparity seen between the training and validation loss indicates the potential for further training to optimize the model’s performance on the validation dataset. The second Figure 5b illustrates that the model trained via data augmentation demonstrated a reduced loss in comparison to the model that was trained. The notable reduction in loss demonstrated how well data augmentation works to support the model’s ability to generalize to new inputs. Furthermore, the model’s loss exhibited a plateau at a reduced level when data augmentation techniques were employed, suggesting enhanced efficacy in the acquisition of knowledge from the training data. In the preliminary phase, both models had encouraging attributes. Model 1 consistently demonstrated an increase in accuracy and suggested the possibility of future improvements with longer training. In the second model, the utilization of data augmentation was observed to result in a noteworthy decrease in loss, hence underscoring the efficacy of these strategies in the context of cancer diagnosis.

#### 4.1.2. Global Model

Table 10 provides an overview of the performance of the global model for the first iteration. The true positive (sensitivity) rate depicts that 97.68% of cancer patients are identified correctly with our proposed model. The correct prediction for patients without cancer also turns out to be good (i.e., 96.77%). On the other hand, low values of false positive and false negative rates establish that the model misclassification of patients with breast cancer and without breast cancer is low. The presented Table indicates an accuracy rate of 97.26%, indicating that the model successfully identified 96.26% of the observations. The precision metric is currently at 96.43%, signifying that 96.43% of the anticipated positive instances were indeed positive. In the current context, it is noteworthy that the recall rate stands at 97.68%, indicating that 97.68% of the true positive instances were accurately classified as such. Significantly, the model demonstrates a proclivity for producing erroneous positives as opposed to false negatives. It implies an increased probability of erroneously categorizing a negative observation as positive. Several aspects can contribute to this phenomenon, such as the quality of the data and the intricacy of the task. Hence, the performance of the model can be considered satisfactory; however, there is room for improvement.

### 4.2. Second Iteration Results

Here are the second iteration’s results.

#### 4.2.1. Local Models

Accuracy: The graph presented in Figure 6a provides a visual representation of the performance trajectory of the ML model throughout 50 trainings (due to early stopping criteria). Upon close examination of the graph, a notable trend becomes apparent. Initially, it is clear that as the number of trainings increases, the model’s accuracy increases steadily and consistently. This indicates that the model can enhance its performance as its familiarity with the training data increases. A crucial observation emerges around 10 when the training accuracy begins to plateau. It suggests that additional training periods may not result in significant gains during the training phase. Significantly, the validation accuracy continues to improve even after 10, eventually surpassing the training accuracy. The difference between training and validation accuracy is reassuring evidence that the model avoids overfitting and is effective at generalizing its knowledge to new, unseen data. The second model, as depicted in Figure 6b, shows an accuracy score of 0.87, indicating that it correctly classified 97.26% of the data (in 70 epochs), which is a commendable accomplishment. Despite the possibility that additional training periods could result in minor enhancements, it is essential to recognize that an accuracy score of 0.97 already indicates a high level of performance. Analyzing the trend of this graph further emphasizes that the model travels from early rapid learning to a phase of stable performance. The disparity between training and validation accuracy demonstrates the model’s ability to generalize effectively and avoid overfitting. This trend demonstrates the efficiency and suitability of the model for the classification task at hand.In both graphs, a similar pattern is observed: an initial phase of rapid accuracy improvement as the models rapidly adapt to the training data. As training continues, however, a transition occurs in which training accuracy stabilizes and reaches a plateau, indicating diminishing returns from additionals. The behavior of validation accuracy distinguishes these models. In both cases, validation accuracy continues to increase even after training accuracy reaches a plateau, demonstrating the models’ ability to generalize to new, unknown information. This disparity between training and validation accuracy highlights the importance of avoiding overfitting, a vital performance indicator. Both models obtained high final accuracy scores, reaching 0.97%, demonstrating their data classification proficiency. These trends collectively illustrate the learning dynamics of the models, demonstrating their ability to balance learning from the training data with effective generalization, thereby making them appropriate for their respective classification tasks.Loss: Upon careful examination of Figure 7a, a discernible trend becomes apparent. Initially, it is notable that as the model undergoes more training, its loss decreases consistently. This trend is extremely favorable, indicating that the model is able to adapt and improve its predictions over time. At 20, when the training loss begins to level off and reaches a plateau, a crucial observation is made. This phenomenon suggests that additional training periods may produce diminishing gains during the phase of training. In contrast, the validation loss continues to decrease until 40, exhibiting a distinct behavior. The difference between training and validation loss is a positive indicator that the model effectively prevents overfitting the training data as it can generalize its acquired knowledge to new, unobserved data with minimal error. The second graph shown in Figure 7b concludes with an ultimate loss score of 0.06, demonstrating the model’s ability to make highly accurate predictions with a small error rate. This outstanding final loss score demonstrates the model’s skill in its predictive assignment.These graphs illustrate how both ML models exhibit a distinct pattern: initial phases of rapid loss reduction as they rapidly adapt to training data. Both models reach a plateau in training loss around 10, indicating decreasing returns from additional trainings. However, the validation loss continues to decrease, demonstrating the models’ capacity to generalize knowledge to new, unobserved data without overfitting. The disparity between training loss and validation loss is a defining characteristic that demonstrates their balanced learning dynamics. While the final loss scores for the graphs are 0.06 and 0.5, respectively, they both indicate the models’ ability to make accurate predictions with varying degrees of precision. These tendencies demonstrate the models’ aptitude for data-driven learning, effective adaptation, and sustaining generalization capabilities while avoiding overfitting.

#### 4.2.2. Global Model

Table 11 depicts the effectiveness of the global model with a 98% correctly classified (i.e., true positive rate) cancer patient. The correct prediction for patients without cancer also turns out to be good (i.e., 96.77%). In addition, a 3.23% false positive rate and a 1.89% false negative rate establish that the model misclassification of patients with breast cancer and without breast cancer is low. Upon comparing the present iteration with our findings of the first iteration, it is evident that in the second iteration, the model has shown improvement in its accuracy level, reaching 98%. This observation indicates the model’s consistent and reliable prediction capability. Moreover, the false negative rate decreased by 0.43% in the second iteration. This demonstrates how the model has performed better by showing a decline in the suggestion of a cancer patient as a negative example. Significantly, the precision is 96.49%, suggesting a greater proportion of accurate positive predictions among the overall positive predictions. Based on these data, it is clear that the model’s performance remains reasonably good. However, there is still potential for additional refining and enhancement. In order to further improve the classification capabilities, feasible strategies may involve investigating other classification algorithms or optimizing the parameters of the existing algorithm. These modifications have the potential to provide a more optimal trade-off between precision and recall, hence resulting in a classification model that is more resilient and dependable.

### 4.3. Third Iteration Results

The results of the third iteration are as follows.

#### 4.3.1. Local Models

Accuracy: In the third iteration of our study, we proceed to assess the efficacy of a discussion of the performance of both models as depicted through a graphical depiction. The provided graphs depict the chronological progression of accuracy in the models, as mentioned earlier, offering significant insights into the models’ performance on both the training and validation datasets. The *x*-axis denotes the number of trainings, while the *y*-axis illustrates the accuracy, providing a lucid depiction of the model’s learning progression.In the first model depicted in Figure 8a, the evaluation of accuracy is determined by calculating the proportion of accurately classified predictions. At the outset, the training accuracy exhibits a notable elevation, although it progressively diminishes due to the model’s tendency to overfit the training data. In contrast, the validation accuracy demonstrates an initial lower value followed by a gradual increase, indicating a significant improvement in the model’s ability to generalize effectively to new and unseen data. Ideally, the convergence of training and validation accuracies indicates a balanced model. In the present scenario, convergence is observed but accompanied by a minor overfitting concern, as indicated by the training accuracy of 94% and the validation accuracy of 97% in 70 epochs.In the second graph, as depicted in Figure 8b, the training accuracy demonstrates convergence at approximately 90%. However, the validation accuracy exhibits convergence at approximately 95%. This indicates that the model exhibits good performance in the training and validation data and remains a commendable model in its entirety. The graph additionally illustrates that the accuracy of the model reaches a peak after approximately 40 epochs (which is due to early stopping criteria). This implies that the model’s learning capacity diminishes beyond this juncture with respect to the training data. The potential exists for enhancing the model by augmenting the number of s. Both models exhibit robust performance with high levels of accuracy.Loss: The present findings entail an analysis of the performance of both models in the third iteration through the examination of their loss curves throughout the training process. In the beginning stages of the first model, as shown in Figure 9a, the training loss exhibits a relatively large value, which after that diminishes gradually as the model progressively adapts to represent the training data accurately. Concurrently, the initial validation loss exhibits a higher value, which then diminishes as the model progressively acquires the ability to generalize to novel data. It is crucial to emphasize that the convergence of both training and validation losses indicates that the model effectively mitigates the risk of overfitting the training data. The training loss demonstrates stability at a value of roughly 0.16, whereas the validation loss remains rather constant at around 0.1. The findings presented in this study clearly indicate that the model exhibits remarkable performance, as evidenced by its low loss metric, hence indicating its efficacy.In the case of the second model, as shown in Figure 9b, it is observed that the training loss exhibits a significant drop early on, indicating its ability to acquire essential data patterns. Nevertheless, it is worth noting that the loss curve reaches a plateau after approximately 20 epochs, suggesting that there is a decrease in the amount of knowledge gained from the training data. In the interim, it is observed that the validation loss demonstrates an initial decrease, followed by an upward trend after about 20 epochs. In an optimal situation, the training and validation losses would exhibit convergence, indicating a model that avoids both overfitting and underperformance. Nevertheless, in this particular instance, the model demonstrates certain tendencies towards overfitting. Although the model exhibits a robust foundation, the potential exists for enhancement through the exploration of tactics such as the reduction in training epochs or the adoption of alternative regularization techniques. Hence, it is concluded that both models demonstrate a notable reduction in loss values, hence confirming their efficacy.

#### 4.3.2. Global Model

During the concluding stage of our model evaluation, we conduct a comprehensive assessment of its performance depicted in Table 12. It presents the effectiveness of the global model in the third and final iteration. There are 98% correctly classified (i.e., true positive rate) cancer patients. The correct prediction rate for patients without cancer also turns out to be good (i.e., 96.77%), similar to previous evaluations of the model. Similarly, a 3.23% false positive rate and a 1.89% false negative rate provide empirical support for the accurate predictions by our model. These results exemplify the proficiency of the model in accurately recognizing and classifying.

The recall statistic, which quantifies the proportion of accurately predicted positive instances relative to the overall count of real positive instances, is approximately 98.0%. This finding indicates that the model exhibited a significant level of precision in effectively capturing and classifying about 98.0% of the positive cases. Nevertheless, it is important to acknowledge that occurrences of misclassifications exist. The presence of 3.32% false positives suggests that our model exhibited misclassification by identifying the events as positive when, in reality, they were negative. This phenomenon adversely impacts the model’s accuracy by encompassing instances where the model’s predictions for positive outcomes were determined to be erroneous. Furthermore, the occurrence of false negative instances suggests that our model failed to correctly identify 1.89% true positive cases, incorrectly categorizing them as negative. It has a negative impact on the recall metric since it implies that there are instances where positive cases were not detected.

Therefore, our model demonstrates a significant degree of performance that includes a significant percentage of $TR_{ps}$ as well as $TR_{ng}$, consequently leading to favorable precision (i.e., 96.5%), accuracy (i.e., 97.54%), and recall (i.e., 98.0%). The presence of a finite number of $FL_{ps}$ as well as $FL_{ng}$ cases suggests the possibility of making improvements to reduce these misclassifications, hence enhancing the reliability and precision of the model.

### 4.4. Overall Results

Figure 10 illustrates the progressive evolution of our global breast cancer detection model’s performance through numerical metrics. It displays the accuracy, precision, recall, and F1-Score across three iterations. Notably, the graph reveals a steady improvement in the model’s capabilities over time. In the first iteration, all performance indicators, such as precision, recall, precision, and F1-Score, exhibit considerably lower values. However, as the model undergoes refinement, which is particularly evident in the third iteration, there is a remarkable enhancement across all these crucial evaluation criteria. Of particular significance is the F1-Score, a comprehensive measure of the model’s overall performance. It exhibits a notable ascent from 97% in the first iteration to a robust 97.24% in the third iteration. This substantial increase indicates that the model has become significantly more accurate, precise, and efficient at capturing relevant instances while minimizing false negatives. Overall, the graph serves as a promising indicator of the model’s potential. It vividly illustrates the model’s progressive improvement, underlining its increasing effectiveness in accurately detecting breast cancer cases based on numerical data. This upward trend instills confidence in the model’s ability to continually enhance its diagnostic capabilities, ultimately benefiting medical applications and patient care.

### 4.5. Comparison with State of the Art Studies

The following table provides a detailed comparison between our study and other research endeavors in the domain of breast cancer detection and diagnosis. We conducted comparisons with existing works that utilized similar datasets or methodologies like federated and deep learning. While there might be variations in specific datasets, we ensured that the compared works shared similarities in terms of the nature and characteristics of the data. The results presented in Table 13 exhibit exceptional performance in various important criteria, including accuracy, precision, recall, and F1-Score. Significantly, our methodology integrates Federated Learning (FL) and Deep Neural Networks (DNN), resulting in an accuracy rate of 97%, precision of 96%, recall of 98.0%, and an exceptional F1-Score of 97%. The noteworthy outcomes highlight the capacity of our technology to potentially transform breast cancer diagnosis while simultaneously prioritizing data privacy and fostering collaboration among healthcare institutions. Additional research and the possibility of incorporating these findings into clinical applications should be pursued.

The methodology employed in our study offers a novel and structured approach to showcasing the outcomes, successfully portraying the progressive development of our research. By conducting a thorough examination of each version, we aim to clarify the incremental process of improvement, thereby providing insights into the ongoing development of our breast cancer detection model. This approach reveals the hierarchical structure of improvements and emphasizes the creative nature of our methods in communicating the advancements made in our work. The final iteration of our project demonstrates the effectiveness of our approach as it exhibits exceptional levels of precision, precision, and recall scores for the detection of breast cancer. The utilization of this innovative approach to presenting information exhibits considerable potential in furthering the development of medical AI applications [22].

## 5. Limitations

The data set utilized in this study exhibits a relatively modest size and is confined to a specific dataset. The constrained variability of the dataset has imposed limitations. Furthermore, the findings presented in this study demonstrate encouraging outcomes in terms of accuracy, along with other performance metrics. However, it is imperative to emphasize the significance of clinical validation in real-world testing as an essential stage in the research process.

## 6. Conclusions

Traditional methods of breast cancer screening often have high false positive rates, leading to unnecessary stress and medical procedures for patients. Additionally, they sometimes fail to detect cancer at its earliest stages when it is most treatable. In this context, AI offers a promising solution. It is crucial to recognize AI’s role in breast cancer diagnosis as an ever-evolving area that requires constant and in-depth exploration to comprehend its potential and limitations fully. New advancements in AI ultimately lead to improved patient outcomes. This study illustrates a significant leap in breast cancer diagnosis by employing an iterative approach to collaborative learning and utilizing DNN-based Federated Learning. It successfully amalgamates the collective knowledge and data resources from various healthcare institutions while ensuring patient privacy and data security measures. It paved the way for more effective early detection and treatment of breast cancer. However, it also highlights the importance of continuous exploration and innovation in this area to fully leverage the potential of these advanced technologies for improving patient care. In the future, we aim to work on the diagnosis of other cancer types, such as skin cancer.

## Figures and Tables

**Figure 1 healthcare-11-03185-f001:**
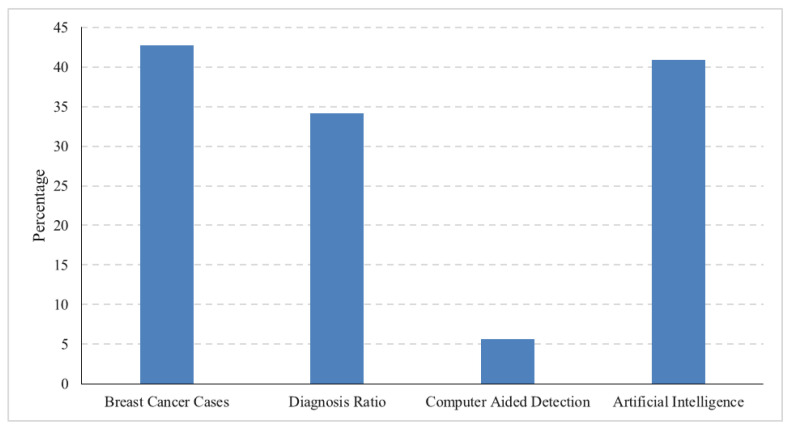
Average ratio among breast cancer cases and detection mechanisms used.

**Figure 2 healthcare-11-03185-f002:**
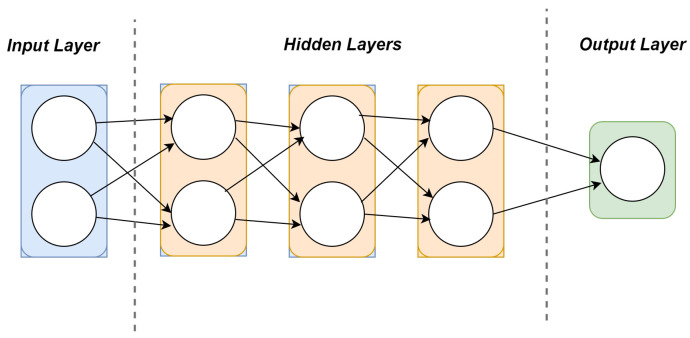
A deep neural network.

**Figure 3 healthcare-11-03185-f003:**
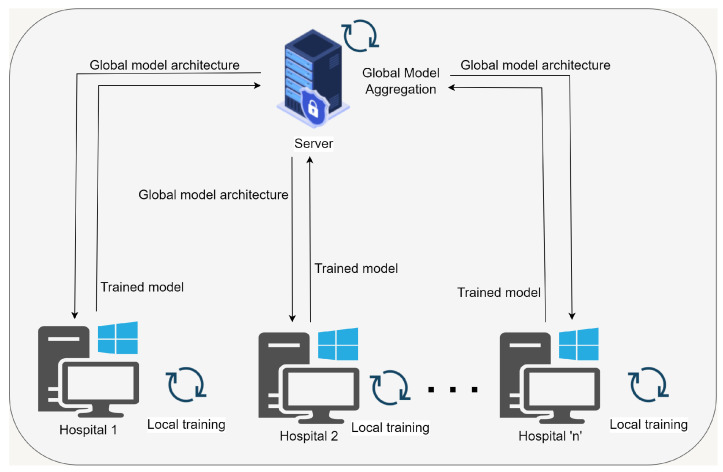
Federated learning.

**Figure 4 healthcare-11-03185-f004:**
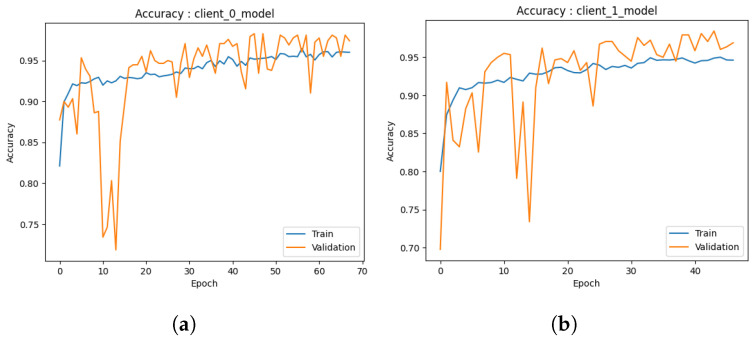
Accuracy comparison in the first iteration. (**a**) Accuracy for first model; (**b**) Accuracy for second model.

**Figure 5 healthcare-11-03185-f005:**
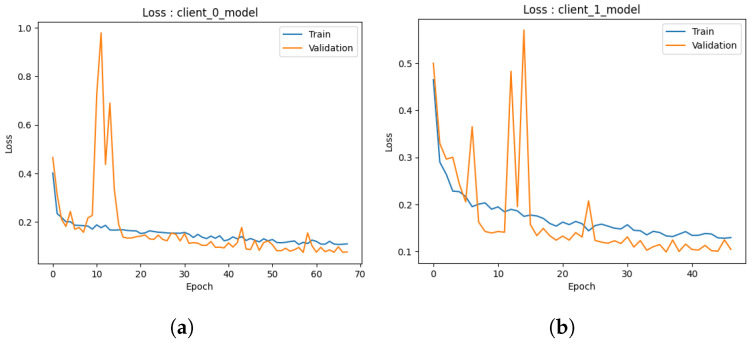
Loss comparison in the first iteration. (**a**) Loss for Model 1; (**b**) loss for Model 2.

**Figure 6 healthcare-11-03185-f006:**
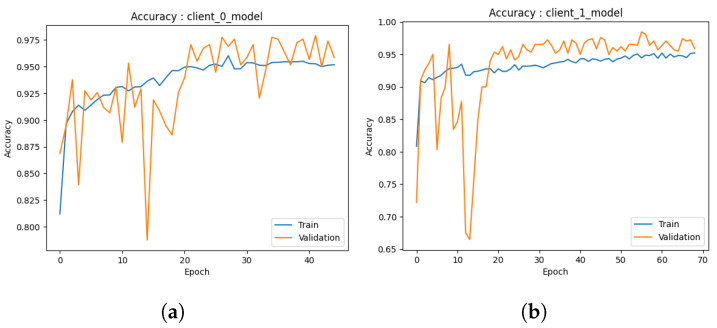
Accuracy comparison in the second iteration. (**a**) Accuracy for Model 1; (**b**) accuracy for Model 2.

**Figure 7 healthcare-11-03185-f007:**
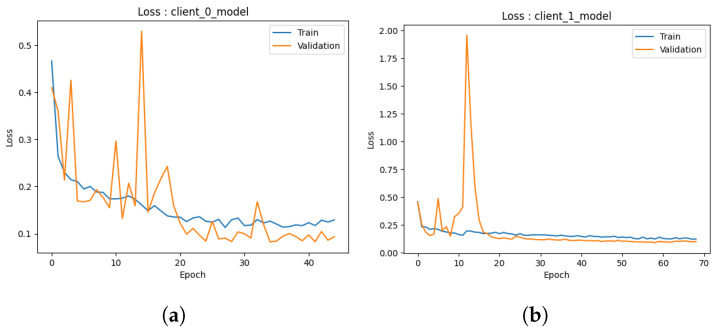
Loss comparison in the second iteration. (**a**) Loss for Model 1; (**b**) loss for Model 2.

**Figure 8 healthcare-11-03185-f008:**
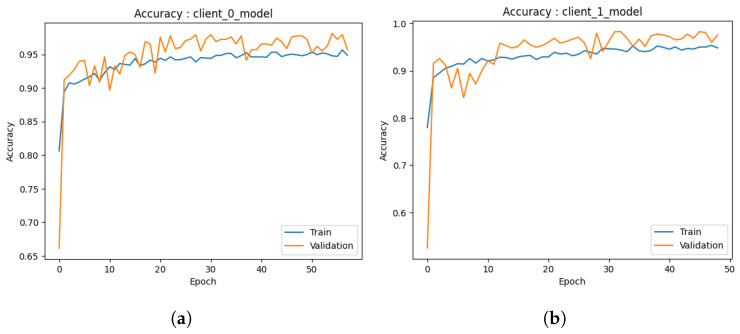
Accuracy comparison in the third iteration. (**a**) Accuracy for Model 1; (**b**) accuracy for Model 2.

**Figure 9 healthcare-11-03185-f009:**
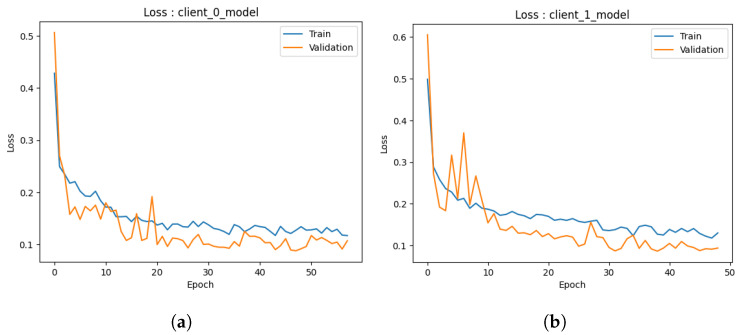
Loss comparison in the third iteration. (**a**) Loss for Model 1; (**b**) loss for Model 2.

**Figure 10 healthcare-11-03185-f010:**
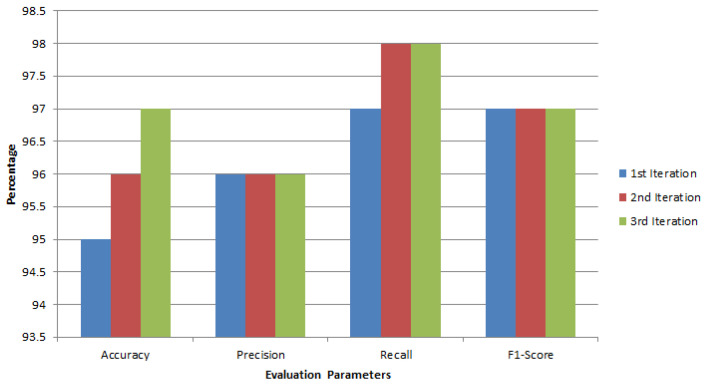
The overall results.

**Table 1 healthcare-11-03185-t001:** Comparison of state-of-the-art studies on breast cancer detection and federated learning.

Ref.	Methodology	Dataset Used	Key Findings	Limitations
[7]	Auto-encoder for unsupervised breast cancer identification	Breast Cancer Wisconsin (Diagnostic) Dataset	Auto-encoder achieved 98.4% precision and recall	No comparison with other unsupervised methods, limited discussion on scalability
[30]	FL framework with CNN	Multiple cancer datasets	Achieved above 90% accuracy for five cancer types	Limited comparison with MLP and data imbalance not addressed
[31]	FL with transfer learning, SMOTE, and FeAvg-CNN + MobileNet model	Mammography datasets	Proposed method showed superior classification performance	No exploration of federated privacy concerns
[25]	FL for histopathology-based breast cancer classification	BreakHis dataset	FL model performed on par with centralized learning	Limited discussion on scalability to larger datasets
[32]	DL-based Xception model	Kaggle dataset	Model exhibited proficiency in breast cancer detection	Increased processing time not discussed
[33]	FL for predicting triple-negative breast cancer	Confidential patient data	FL enhanced the effectiveness of detecting triple cancer	Specific FL implementation details not provided, potential variability in data quality

**Table 2 healthcare-11-03185-t002:** Dataset details.

Attribute	Description
Number of Institutions	University of Wisconsin Hospital in Madison, Madison, WI, USA
Data Types	Demographic, Clinical, Pathological, Outcome
Data Formats	Numerical, Categorical, Images
Patient Data Variations	Demographic, Clinical, Pathological, Outcome

**Table 3 healthcare-11-03185-t003:** Patient data details.

Data Type	Specific Data Element	Patient Data Variations
Demographic	Age	Range from 25 to 97 years old
	Sex	Male or female
	Race	White, Black, Asian, or other
	Ethnicity	Hispanic, Non-Hispanic
Clinical	Medical History	Presence or absence of various medical conditions
	Medications	Types and dosages of medications
	Laboratory Results	Variations in blood cell counts, hormone levels, and other biomarkers
	Imaging Studies	Differences in tumor size, shape, and texture on mammograms, ultrasounds, or other imaging modalities
Pathological	Tumor Pathology	Type of breast cancer (e.g., ductal carcinoma, lobular carcinoma)
	Molecular Profiling	Genetic alterations in tumor cells
Outcome	Survival	Time from diagnosis to death or censoring
	Recurrence	Presence or absence of tumor recurrence after treatment
	Treatment Response	Tumor shrinkage or growth in response to treatment

**Table 4 healthcare-11-03185-t004:** Features of BCW (diagnostic) dataset.

Feature	Description
id	Identifier
concave_points_worst	Worst concave points on the perimeter of nuclei
diagnosis	Diagnosis (B = benign, M = malignant)
symmetry_worst	Worst symmetry of nuclei
radius_mean	Mean radius of nuclei
fractal_dimension_worst	Worst fractal dimension of nuclei
texture_mean	Mean texture of nuclei
texture_se	Standard error of texture
perimeter_mean	Mean perimeter of nuclei
perimeter_se	Standard error of perimeter
area_mean	Mean area of nuclei
area_se	Standard error of area
smoothness_mean	Mean smoothness (1–3) of nuclei
smoothness_worst	Worst smoothness (1–3) of nuclei
compactness_mean	Mean compactness of nuclei
compactness_worst	Worst compactness of nuclei
concavity_mean	Mean concavity of nuclei
concavity_worst	Worst concavity of nuclei
concave_points_mean	Mean of concave points on the perimeter of nuclei
concave_points_worst	Worst of concave points on the perimeter of nuclei
symmetry_mean	Mean symmetry of nuclei
symmetry_worst	Worst symmetry of nuclei
fractal_dimension_mean	Mean fractal dimension of nuclei
fractal_dimension_worst	Worst fractal dimension of nuclei
radius_se	Standard error of radius

**Table 5 healthcare-11-03185-t005:** Dataset preprocessing summary.

Preprocessing Step	Dataset Distribution
Original dataset distribution	569
Benign (B)	357
Malignant (M)	212
After outliers removal	538
Benign (B)	340
Malignant (M)	198
Data Augmentation	
Adding noise (0 to 0.01)	Total = 5742
Benign (B)	3212
Malignant (M)	2530
SMOTE for class balance	Total = 6424
Benign (B)	3212
Malignant (M)	3212

**Table 6 healthcare-11-03185-t006:** Interquartile range (IQR).

Feature	IQR	Feature	IQR
radius_mean	5.42	texture_mean	5.29
perimeter_mean	36.72	area_mean	504.1
smoothness_mean	0.02	compactness_mean	0.07
concavity_mean	0.12	concave points_mean	0.06
symmetry_mean	0.03	fractal_dimension_mean	0.01
radius_se	0.3	texture_se	0.58
perimeter_se	2.03	area_se	39.02
smoothness_se	0.0	compactness_se	0.02
concavity_se	0.03	concave points_se	0.01
symmetry_se	0.01	fractal_dimension_se	0.0
radius_worst	7.45	texture_worst	8.56
perimeter_worst	52.31	area_worst	766.36
smoothness_worst	0.03	compactness_worst	0.21
concavity_worst	0.3	concave points_worst	0.11
symmetry_worst	0.07	fractal_dimension_worst	0.02

**Table 7 healthcare-11-03185-t007:** Dataset before data augmentation.

Features	Mean	Std	Min	25%	50%	75%	Max
radius_mean	14.46	3.1	6.98	12.19	13.73	16.75	23.27
texture_mean	19.07	3.74	9.71	16.39	18.92	21.47	29.81
perimeter_mean	93.9	21.12	43.79	78.2	88.48	109.72	152.1
area_mean	675.59	294.56	143.5	458.02	581.65	884.78	1686.0
smoothness_mean	0.1	0.01	0.06	0.09	0.1	0.1	0.13
compactness_mean	0.1	0.04	0.02	0.07	0.09	0.13	0.23
concavity_mean	0.08	0.06	0.0	0.03	0.06	0.12	0.28
concave points_mean	0.05	0.03	0.0	0.02	0.04	0.08	0.16
symmetry_mean	0.18	0.02	0.12	0.16	0.18	0.19	0.25
fractal_dimension_mean	0.06	0.01	0.05	0.06	0.06	0.06	0.08
radius_se	0.37	0.18	0.11	0.23	0.32	0.47	0.97
texture_se	1.09	0.38	0.36	0.82	1.04	1.34	2.24
perimeter_se	2.59	1.24	0.76	1.59	2.28	3.22	6.79
area_se	36.37	25.4	6.8	18.15	26.38	47.48	116.29
smoothness_se	0.01	0.0	0.0	0.01	0.01	0.01	0.01
compactness_se	0.02	0.01	0.0	0.01	0.02	0.03	0.06
concavity_se	0.03	0.01	0.0	0.02	0.02	0.04	0.08
concave points_se	0.01	0.0	0.0	0.01	0.01	0.01	0.02
symmetry_se	0.02	0.01	0.01	0.01	0.02	0.02	0.03
fractal_dimension_se	0.0	0.0	0.0	0.0	0.0	0.0	0.01
radius_worst	16.72	4.27	7.93	13.35	15.64	19.93	28.01
texture_worst	25.66	5.47	12.02	21.71	25.62	29.88	42.65
perimeter_worst	109.71	28.99	50.41	86.77	102.4	131.42	187.33
area_worst	911.11	472.97	185.2	549.28	749.5	1227.85	2405.48
smoothness_worst	0.13	0.02	0.08	0.12	0.13	0.15	0.19
compactness_worst	0.25	0.12	0.03	0.16	0.22	0.32	0.66
concavity_worst	0.27	0.17	0.0	0.13	0.24	0.39	0.77
concave points_worst	0.12	0.06	0.0	0.07	0.11	0.16	0.26
symmetry_worst	0.29	0.05	0.18	0.26	0.29	0.32	0.43
fractal_dimension_worst	0.08	0.01	0.06	0.07	0.08	0.09	0.12
diagnosis	0.44	0.5	0.0	0.0	0.0	1.0	1.0

**Table 8 healthcare-11-03185-t008:** Dataset after data augmentation.

Feature	Mean	Std	Min	25%	50%	75%	Max
radius_mean	14.46	3.1	6.95	12.19	13.72	16.78	23.28
texture_mean	19.06	3.73	9.7	16.38	18.92	21.47	29.82
perimeter_mean	93.9	21.1	43.78	78.18	88.48	109.73	152.12
area_mean	675.59	294.3	143.49	457.9	581.65	886.3	1686.02
smoothness_mean	0.1	0.01	0.05	0.09	0.1	0.11	0.15
compactness_mean	0.1	0.04	0.0	0.07	0.09	0.13	0.25
concavity_mean	0.08	0.06	−0.01	0.03	0.07	0.12	0.31
concave points_mean	0.05	0.03	−0.01	0.02	0.04	0.08	0.18
symmetry_mean	0.17	0.03	0.09	0.16	0.17	0.19	0.25
fractal_dimension_mean	0.06	0.01	0.04	0.06	0.06	0.07	0.11
radius_se	0.37	0.18	0.1	0.23	0.32	0.47	0.98
texture_se	1.09	0.38	0.35	0.81	1.04	1.34	2.25
perimeter_se	2.59	1.24	0.74	1.58	2.27	3.22	6.81
area_se	36.37	25.38	6.78	18.15	26.39	47.59	116.31
smoothness_se	0.01	0.01	−0.01	0.0	0.01	0.01	0.03
compactness_se	0.02	0.02	−0.03	0.01	0.02	0.03	0.08
concavity_se	0.03	0.02	−0.01	0.01	0.02	0.04	0.1
concave points_se	0.01	0.01	−0.01	0.01	0.01	0.02	0.06
symmetry_se	0.02	0.01	−0.01	0.01	0.02	0.02	0.05
fractal_dimension_se	0.0	0.01	−0.02	−0.01	0.0	0.01	0.03
radius_worst	16.72	4.27	7.91	13.35	15.64	19.93	28.04
texture_worst	25.66	5.47	12.0	21.7	25.62	29.89	42.66
perimeter_worst	109.71	28.97	50.39	86.76	102.4	131.44	187.34
area_worst	911.11	472.56	185.19	549.1	749.5	1228.0	2405.5
smoothness_worst	0.13	0.02	0.07	0.12	0.13	0.15	0.2
compactness_worst	0.25	0.12	0.01	0.16	0.22	0.33	0.67
concavity_worst	0.27	0.17	−0.01	0.13	0.25	0.38	0.78
concave points_worst	0.11	0.06	−0.02	0.07	0.11	0.16	0.28
symmetry_worst	0.29	0.05	0.16	0.25	0.28	0.32	0.44
fractal_dimension_worst	0.08	0.02	0.03	0.07	0.08	0.09	0.13
diagnosis	0.44	0.5	0.0	0.0	0.0	1.0	1.0

**Table 9 healthcare-11-03185-t009:** Classification outcomes for breast cancer prediction.

Outcome	Description
$TR_{ps}$	Model correctly predicts breast cancer in a patient with breast cancer
$TR_{ng}$	Model correctly predicts no breast cancer in a patient without breast cancer
$FL_{ps}$	Model incorrectly predicts breast cancer in a patient without breast cancer
$FL_{ng}$	Model incorrectly predicts no breast cancer in a patient with breast cancer

**Table 10 healthcare-11-03185-t010:** Global model performance metrics for the first iteration.

Metric	Value
True Positive Rate (Sensitivity)	97.68%
False Positive Rate	3.23%
True Negative Rate (Specificity)	96.77%
False Negative Rate	2.32%
Accuracy	97.26%
Precision	96.43%
F1-Score	97.05%

**Table 11 healthcare-11-03185-t011:** Global model performance metrics for the second iteration.

Metric	Value
True Positive Rate (Sensitivity)	98.00%
False Positive Rate	3.23%
True Negative Rate (Specificity)	96.77%
False Negative Rate	1.89%
Accuracy	97.54%
Precision	96.49%
F1-Score	97.24%

**Table 12 healthcare-11-03185-t012:** Global model performance metrics for the third iteration.

Metric	Value
True Positive Rate (Sensitivity)	98.00%
False Positive Rate	3.23%
True Negative Rate (Specificity)	96.77%
False Negative Rate	1.89%
Accuracy	97.54%
Precision	96.49%
F1-Score	97.24%

**Table 13 healthcare-11-03185-t013:** Comparison of breast cancer detection results.

Study	Model	Techniques	Results
Proposed Study	DNN (FL)	Federated Learning	Accuracy: 97.5%, Precision: 96%, Recall: 98.0%, and F1-Score: 97%
[7]	Auto-encoder	Unsupervised	Precision: 98.4%
[30]	Hybrid FL with CNN	Transfer Learning	Accuracy: Above 90%
[36]	DTC	Feature Selection	Accuracy: 94
[25]	FCN	Machine Learning	Accuracy: 96.4%, Sensitivity: 97.5%, and Specificity: 97.8%.
[47]	CNN	Machine Learning	Accuracy 99.7%

## Data Availability

Can be obtained from author 2 using noshina.tariq@mail.au.edu.pk.

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
