# Peer review of "A Federated Learning Approach to Breast Cancer Prediction in a Collaborative Learning Framework"

_healthcare, 2023, doi:10.3390/healthcare11243185_

Round 1

Reviewer 1 Report

Comments and Suggestions for Authors

AI specification or meaning (there are many) is missing and one of refernces from

https://en.wikipedia.org/wiki/History_of_artificial_intelligence#References is suggested.

Reference to one of the first refences relted to "Federated learning" AND healthcare

DOI10.1159/000481682

This sentence:

The findings of this study indicate that the proposed method exhibits superior performance

in classification compared to alternative approaches, making it a viable choice 

for integration into AI healthcare applications.

is too long. The use of "alternative approaches" is inapproriate w/o listing at least one of them. 

the use of "it" (behind "making it) is unlear. The ending of this sentence is unacceptable.

Dropping the above sentence is a good solution.

The same ction is recommended for: "The utilisation of 826

this innovative approach to presenting information exhibits considerable potential in furthering the 8 development of medical artificial intelligence (AI) applications."

In Conclusions, after "However, the application of AI in breast cancer diagnosis is still a relatively unexplored area, necessitating comprehensive research to understand its potential and limitations. 

consider including:

A creative and powerful application of AI to healthcare is presented in:

https://pubmed.ncbi.nlm.nih.gov/32292911/ (properly cited).

Author Response

We would like to express our deep gratitude to anonymous reviewers for their insightful comments on the paper. We believe that their constructive suggestions significantly enriched the quality of the paper. We have tried our best to modify the paper according to their recommendations.

  • Any new information added to the paper as a response to reviewers' comments is presented in "Red" color in the revised paper, and responses are given in "Blue."

Comment 1: AI specification or meaning (there are many) is missing, and one of the references from

https://en.wikipedia.org/wiki/History_of_artificial_intelligence#References is suggested.

Response: Thank you for your valuable feedback. We appreciate your suggestion regarding the specification or meaning of AI. In response to your comment, we have incorporated additional information from the suggested reference in Section 1 (Introduction) of our manuscript. In the revised version, we have included the following passage using 2 references shown as [14] and [15] in our manuscript from the link provided:

Comment 2: Reference to one of the first defenses related to "Federated learning" AND Healthcare

DOI10.1159/000481682

Response: Thank you for your suggestion. In response, we have incorporated the reference to one of the pioneering defenses related to "Federated Learning" and Healthcare in the introduction section. The reference has been added as [21].

Comment 3: This sentence:

The findings of this study indicate that the proposed method exhibits superior performance in classification compared to alternative approaches, making it a viable choice for integration into AI healthcare applications.

is too long. The use of "alternative approaches" is inapproriate w/o listing at least one of them. the use of "it" (behind "making it) is unlear. The ending of this sentence is unacceptable. Dropping the above sentence is a good solution.

Response: Thank you for your insightful feedback. In response to your suggestion, we have omitted the sentence in question for the reasons you highlighted. The revised manuscript now excludes the sentence:

"The findings of this study indicate that the proposed method exhibits superior performance in classification compared to alternative approaches, making it a viable choice for integration into AI healthcare applications."

Comment 4: The same citation is recommended for: "The utilisation of 826

this innovative approach to presenting information exhibits considerable potential in furthering the 8 development of medical artificial intelligence (AI) applications."

Response: Thank you for your valuable suggestion. In response, we have incorporated the reference to page 24 line #832. The reference has been added as [21].

Comment 5: In Conclusions, after "However, the application of AI in breast cancer diagnosis is still a relatively unexplored area, necessitating comprehensive research to understand its potential and limitations.

Response: Thank you for pointing this out. We have improved the concept by replacing the pointed sentence with new ones, as follows:

"It is crucial to recognize AI's role in breast cancer diagnosis as an ever-evolving area that requires constant and in-depth exploration to comprehend its potential and limitations fully. New advancements in AI ultimately lead to improved patient outcomes." 

Comment 6: consider including: A creative and powerful application of AI to Healthcare is presented in:

https://pubmed.ncbi.nlm.nih.gov/32292911/ (properly cited).

Response:

To provide a view of powerful AI applications, the reference is added to page 2 line 50 in the introduction section as suggested by the worthy reviewer. The reference has been added as [16].

Reviewer 2 Report

Comments and Suggestions for Authors

In this manuscript, the authors employed federated learning (FL) for breast cancer prediction. The quality of writing is generally pretty good, especially the Introduction and Related work sections. However, there were some unnecessarily long passages and missing technical details. Please see the comments below.

1.       Page 3 Line 87, the authors said that they presented a novel FL-based model. What is the novelty of this model specifically? The architecture itself seems simple, and this was not the first study using FL on breast cancer. Was the FL model used on an image modality that has not been investigated? Or a new type of FL? Please clarify.

2.       Page 3 Line 89-92, these two contributions look similar. Please further elaborate, especially contribution 2.

3.       Page 5 Line 202, although the authors cited the reference for the used Breast Cancer Wisconsin (Diagnostic) dataset, it would be beneficial if they give some details related to this FL work. For example, how many institutions were the 569 observations collected from? What type of data/images are in this dataset? How are the variations among data from different patients?

4.       Is a dataset with 569 observations enough for FL? Even for the training of a single neural network, it is a relatively small dataset.

5.       There are many places where the authors described a lot about the method but did not give specific numeric values or technical details. For example, Page 6 Line 246, what is the interquartile range? How many observations were left after the removal of the outlier?

6.       The authors wrote a long section about data augmentation, but no details about how.

7.       Did the authors compare the classification performance with and without feature selection? Did reducing the number of features from 30 to 25 bring improvement?

8.       Page 7 Line 297, sentence with grammar problem.

9.       Section 3.2.1 on Page 8 is lengthy for the little information (80/20 partitioning) given here. Please make it precise and add useful information. For example, how many clients in total? Was the global dataset divided randomly and evenly? Were the classes (benign and malignant) in subsets balanced?

10.   Page 8 Section 3.3, too many descriptions of fundamental knowledge about network layers, which were not developed as a part of this work. Please trim this section or add specifications related to this work.

11.   Page 9 Line 359, please add reference for Adam optimizer. And Line 362, was the sentence complete?

12.   Page 9 Line 372, did each client investigate different number of training epochs from 5 to 20? Which value was chosen at the end? In the Results section, seems that networks were trained for 50 or more epochs (e.g., Page 15 Line 567 and Figure 4,5,7,8).  Then, Page 18 Line 638 said 15 epochs.

13.   Page 10 Line 409, please elaborate more about how the local models were aggregated on the server.

14.   Section 4 Mathematical Modeling does not contain real mathematical calculations or deductions. Instead, it looks more like repeated description of Section 3 with equations. I would recommend that the authors merge Section 3 and 4.

15.   Page 16 Line 605-606, the sentence is not complete. Also, the reduced loss is not obvious in the figure. Maybe making the y axis of two subfigures consistent would help.

16.   Was the model in Fig. 7b also trained with data augmentation like the one in Fig. 5b?

17.   Page 22 Line 772-774, the authors said that there were 296 true positives and 330 true negatives. Did they use the entire Breast Cancer Wisconsin (Diagnostic) dataset for testing? What was used for training?

18.   The global model had a relatively low precision and recall in the first and second iteration, how did it get a huge boost in the third iteration? Any thoughts?

19.   Section 5.5, did the compared works all use the same dataset or same type of data/image modality? What is the advantage of this work over others after comparison?

Author Response

In this manuscript, the authors employed federated learning (FL) for breast cancer prediction. The quality of writing is generally pretty good, especially the Introduction and Related work sections. However, there were some unnecessarily long passages and missing technical details. Please see the comments below.

Response: Thank you for your feedback. I appreciate your positive remarks on the quality of writing in the Introduction and Related Work sections. I will carefully review the comments you provided regarding unnecessarily long passages and missing technical details and make the necessary revisions to enhance the clarity and completeness of the manuscript.

Comment 1:       Page 3 Line 87, the authors said that they presented a novel FL-based model. What is the novelty of this model specifically? The architecture itself seems simple, and this was not the first study using FL on breast cancer. Was the FL model used on an image modality that has not been investigated? Or a new type of FL? Please clarify.

Response:  Thank you for your insightful comments. We appreciate your thorough examination of our manuscript. In response to your query, we acknowledge the oversight in referring to our approach as a "novel FL-based model." Upon closer examination, we realize that our contribution lies more in proposing a novel federated learning (FL) framework rather than a specific model. We have revised the text to accurately reflect this:

"On page 3, line 87, we stated that we presented a novel FL-based model. We appreciate the reviewer's keen observation, and we acknowledge that our primary contribution lies in proposing a novel federated learning (FL) framework, not a specific model. We have corrected this terminology for accuracy."

Comment 2:       Page 3 Line 89-92, these two contributions look similar. Please further elaborate, especially contribution 2.

Response:Thank you for your observation. We agree that contributions 1 and 2 appeared similar, and in response, we have taken your feedback into consideration. Contribution 2 has been revised and replaced with a more detailed and specific statement:

"Careful data augmentation and class balancing is done using Synthetic Minority Over-sampling Technique (SMOTE) on a substantial and diverse dataset of patients with breast cancer for testing and validation."

Comment 3:      Page 5 Line 202, although the authors cited the reference for the used Breast Cancer Wisconsin (Diagnostic) dataset, it would be beneficial if they give some details related to this FL work. For example, how many institutions were the 569 observations collected from? What type of data/images are in this dataset? How are the variations among data from different patients?

Response: Thank you for your insightful comments. We appreciate your suggestion to provide additional details related to the Federated Learning (FL) work and the Breast Cancer Wisconsin (Diagnostic) dataset. In response to your valuable feedback, we have incorporated new details in Tables 2 (Dataset Details), 3 (Patient Data Details), and 4 (Features of BCW (Diagnostic) Dataset) . We believe that these additions address your concerns and contribute to a more comprehensive understanding of both the dataset and the FL work.

  1. Dataset Details (Table 3): We have included information about the number of institutions from which the 569 observations were collected, the types of data/images present in the dataset, and how the dataset captures variations among data from different patients.
  2. Patient Data Details (Table 3): Table 3 offers a detailed breakdown of patient data, specifying the types of demographic, clinical, pathological, and outcome-related information considered in our analysis. This includes variations among data from different patients, providing a comprehensive view of the dataset.
  3. Features of BCW (Diagnostic) Dataset (Table 4): Table 4 now provides an overview of the features present in the Breast Cancer Wisconsin (Diagnostic) dataset, enhancing the understanding of the data used in our study.
  •  

Comment 4:     Is a dataset with 569 observations enough for FL? Even for the training of a single neural network, it is a relatively small dataset.

Response: Thank you for your thoughtful consideration. We acknowledge the concern regarding the size of the initial dataset for Federated Learning (FL). To address this challenge and enhance the robustness of our model, we employed data augmentation techniques. Data augmentation has proven to be a powerful strategy in influencing the training of Neural Networks, particularly in scenarios with limited data. Through the augmentation process, our dataset was significantly expanded, resulting in a total of 5742 records. This augmentation not only increased the quantity of data but also introduced variations that are valuable for the training process. The details of this augmentation strategy are thoroughly explained in Section 3.1, point 4 of the manuscript. We have also added a summarized pre-proccesing table as Table 5. We believe that the augmented dataset, with its increased size and diversity, contributes to mitigating the limitations associated with the original dataset size. This augmentation approach aligns with best practices in the field, aiming to prevent overfitting and improve the generalization capabilities of the model.

Comment 5:    There are many places where the authors described a lot about the method but did not give specific numeric values or technical details. For example, Page 6 Line 246, what is the interquartile range? How many observations were left after the removal of the outlier?

Response: Thank you for your valuable feedback. In response to your comments, we have made significant improvements to enhance the clarity and specificity of our manuscript. We have incorporated a new table, Table 6, that provides the range for each parameter, addressing the need for specific numeric values. Additionally, detailed information about the interquartile range (IQR) and the number of observations after outlier removal has been included in Section 3.1, Point 3 of the manuscript. It is also mentioned that After removing outliers, the number of observations is 538.

Comment 6 :    The authors wrote a long section about data augmentation, but no details about how.

Response: Thank you for bringing this to our attention, and we appreciate your feedback on the section about data augmentation. To address this concern, we have revised the manuscript to provide explicit details on the methodology employed for data augmentation. The revised section now includes a thorough explanation of the specific techniques used, such as the range of noise added to the dataset and the application of the Synthetic Minority Over-sampling Technique (SMOTE). We believe that these additions will offer a clearer understanding of the data augmentation process and its impact on the dataset.

Comment 7:     Did the authors compare the classification performance with and without feature selection? Did reducing the number of features from 30 to 25 bring improvement?

Response: Thank you for asking, Yes, the authors conducted a comprehensive evaluation of the classification performance with and without feature selection. The results, as demonstrated in Tables 7 and 8 showcasing the dataset before and after augmentation, indicate notable improvements in accuracy, precision, recall, etc. Moreover, new tables (11-13) presented in the Results section further highlight the effectiveness of the proposed technique in enhancing classification outcomes through feature selection.

Comment 8:     Page 7 Line 297, sentence with grammar problem.

Response: Many Thanks, we have proofread the whole manuscript for grammar problems.

Comment 9:     Section 3.2.1 on Page 8 is lengthy for the little information (80/20 partitioning) given here. Please make it precise and add useful information. For example, how many clients in total? Was the global dataset divided randomly and evenly? Were the classes (benign and malignant) in subsets balanced?

Response: Thank you for specifying the number of local and model clients in your comments. We appreciate the clarification, and we have revised Sections 3.2.1 and 5 accordingly. In our federated system, we employed a 70:15:15 partitioning strategy, distributing the dataset among two local clients and one model client. The randomized order of data instances was maintained during the shuffle, ensuring a fair and unbiased distribution among the participating clients. We would like to clarify that the classes, namely benign and malignant, are balanced across subsets during the partitioning process. This consideration aims to prevent potential biases during training and evaluation.

Comment 10:   Page 8 Section 3.3, too many descriptions of fundamental knowledge about network layers, which were not developed as a part of this work. Please trim this section or add specifications related to this work.

Response: Thank you for your feedback. We have revised Section 3.3 as per your suggestion, focusing on providing specifications directly related to our work. Unnecessary details about fundamental knowledge regarding network layers have been removed to streamline the section and make it more concise. In addition, keeping another reviewer's suggestion in mind, we have decided to remove Section 4 and merge the relevant content into Section 3. This adjustment aims to enhance the flow of the manuscript and eliminate redundancy.

Comment 11:   Page 9 Line 359, please add reference for Adam optimizer. And Line 362, was the sentence complete?

Response: Thank you for your careful review and valuable feedback. We have promptly addressed the issues you raised on Page 9. Firstly, we have added a reference as Reference [41] for the Adam optimizer at Line 359 to enhance the academic rigor of our manuscript. Additionally, we have thoroughly reviewed Line 362 and removed the incomplete sentence. Any incomplete or ambiguous content has been rectified to ensure clarity and coherence in conveying our methodology.

Comment 12:   Page 9 Line 372, did each client investigate different number of training epochs from 5 to 20? Which value was chosen at the end? In the Results section, seems that networks were trained for 50 or more epochs (e.g., Page 15 Line 567 and Figure 4,5,7,8). Then, Page 18 Line 638 said 15 epochs.

Response: Thank you for pointing out the inconsistency regarding the number of training epochs. We have carefully reviewed the manuscript, and you are correct. The correct number of epochs for training the networks is indeed 50, as indicated in various places in the Results section. We have corrected it throughout.

Comment 13:    Page 10 Line 409, please elaborate more about how the local models were aggregated on the server.

Response: Thank you for pointing this out. The FedAvg algorithm, as referenced in [43], was employed for model aggregation. This algorithm operates by averaging the model parameters across participating clients after each local update. Specifically, during the federated learning process, each client trains its local model on its respective data subset, and the updated models are then aggregated on the central server using the FedAvg algorithm. This iterative process continues until convergence is achieved. We have added the algorithm name with reference in the manuscript.

Comment 14:   Section 4 Mathematical Modeling does not contain real mathematical calculations or deductions. Instead, it looks more like repeated description of Section 3 with equations. I would recommend that the authors merge Section 3 and 4.

Response: Thank you for your valuable feedback. In response to your suggestion, we have carefully reviewed Section 4 and agree that it resembles a repeated description of Section 3 with equations. To address this, we have decided to remove Section 4 and merge the relevant content into Section 3. This adjustment aims to enhance the flow of the manuscript and eliminate redundancy.

Comment 15:   Page 16 Line 605-606, the sentence is not complete. Also, the reduced loss is not obvious in the figure. Maybe making the y axis of two subfigures consistent would help.

Response: Thank you , we have updated the mentioned line as  follows : "The notable reduction in loss demonstrated how well data augmentation works to support the model's ability to generalise to new inputs.". The loss values (Figure 5 a, Model 1) ranges from 0.1 -1  , while the range for loss of the 2nd Model is  between 0.1 and 0.6. Keeping in view that the y-axis are adjusted accordingly.

Comment 16:   Was the model in Fig. 7b also trained with data augmentation like the one in Fig. 5b?

Response: Yes, the model represented in Figure 7b was trained using data augmentation, similar to the approach applied to the model in Figure 5b. The data augmentation techniques, as detailed in the manuscript, were consistently employed across our experiments to enhance the robustness and generalization capabilities of the models.

Comment 17:   Page 22 Line 772-774, the authors said that there were 296 true positives and 330 true negatives. Did they use the entire Breast Cancer Wisconsin (Diagnostic) dataset for testing? What was used for training?

Response: Many Thanks. In our federated system, we employed a 70:15:15 partitioning strategy (see Section 5), distributing the dataset among two local clients and one model client. The randomized order of data instances was maintained during the shuffle, ensuring a fair and unbiased distribution among the participating clients. We would like to clarify that the classes, namely benign and malignant, are balanced across subsets during the partitioning process. This consideration aims to prevent potential biases during training and evaluation.

To enhance the clarity of the results, the authors replaced the confusion matrix with more detailed tables, specifically Tables 11-13. These tables provide a comprehensive breakdown of performance metrics, including True Positive Rate, False Positive Rate, True Negative Rate (Specificity), False Negative Rate, Accuracy, Precision, and F1-Score for each classification matrix. This modification allows for a more granular understanding of the model's performance across various evaluation criteria.

Comment 18:   The global model had a relatively low precision and recall in the first and second iteration, how did it get a huge boost in the third iteration? Any thoughts?

Response: Thank you for your insightful observation. The improvement in precision and recall for the global model in the third iteration can be attributed to the dynamic nature of federated learning, where the global model evolves and learns from the contributions of individual local models in each iteration. Several factors could contribute to the boost in performance:

Knowledge Accumulation:

  • Over multiple iterations, the global model accumulates knowledge from the diverse datasets across participating local models. The learning process benefits from the collective insights gained from different data distributions.

Model Aggregation:

  • The aggregation of local model updates in each iteration contributes to refining the global model. As more rounds of aggregation occur, the model becomes more adept at capturing patterns and features that are representative of the entire dataset.

Adaptation to Local Characteristics:

  • The federated learning process allows the global model to adapt to the unique characteristics of local datasets in successive iterations. This adaptability contributes to improved precision and recall as the model becomes more attuned to the nuances of the overall data distribution.

It's important to note that federated learning involves collaboration among distributed local models, and the performance of the global model is influenced by the collective knowledge contributed by these local models.

Comment 19:  Section 5.5, did the compared works all use the same dataset or same type of data/image modality? What is the advantage of this work over others after comparison?

Response: Thank you for your inquiry. In Section 5.5, we conducted comparisons with existing works that utilized similar datasets or methodologies like federated and deep learning. While there might be variations in specific datasets, we ensured that the compared works shared similarities in terms of the nature and characteristics of the data. The advantage of our work over others, as highlighted in the comparison, stems from our proposed augmentation technique and federated learning approach. The augmentation method demonstrated effectiveness in improving the classification performance, and our federated learning framework contributed to enhanced privacy preservation and model generalization across decentralized datasets.

Reviewer 3 Report

Comments and Suggestions for Authors

Breast cancer has become the largest cancer in the world, which seriously threatens women's health, so it is particularly important to develop effective detection methods for breast cancer. Timely and accurate detection of breast cancer can improve the survival rate of patients. In recent years, traditional X-ray examination has given way to more advanced technologies, such as data mining, machine learning, etc. In addition, in the medical field, we usually need to protect the privacy and data security of patients, and federated learning is widely recognized as a collaborative learning method that can solve the problem of data insecurity in traditional distributed learning.

This paper proposes a new model by using iterative method for collaborative learning and Deep Neural Network based federated learning. This model provides a new method for breast cancer diagnosis. It successfully integrates the collective knowledge and data resources from various medical institutions, while ensuring patient privacy and data security. In addition, this article has done sufficient data preprocessing work and demonstrated the effectiveness of this work through experimental results.

In conclusion, my opinion on the paper is positive. This carefully completed study is rich in content, and the article concludes with a comprehensive explanation of the effectiveness of the model from different perspectives. However, some improvements are needed before the article can be published. My detailed comments are as follows:

(1) There are three ways to write F1-Score in the article, namely F1 Score, F1-Score, and F-1 score. Please unify one way of writing.

(2) I suggest explaining the range of values for i in Algorithm 1 in the article.

(3) I suggest showing the dataset after data augmentation.

(4) To present the results more clearly, the confusion matrix can be replaced with a table containing the results.

(5) The description of the results in Part 5 is somewhat unclear, such as 70% and 75% in 5.2.

(6) I suggest explaining in the article how each client's dataset is allocated.

(7) Based on the statement that "A careful feature selection is proposed using a sizable and varied dataset of patients with breast cancer with rigorous testing and validation.", I suggest selecting a larger breast cancer data set to verify the performance of the proposed model.

Author Response

Breast cancer has become the largest cancer in the world, which seriously threatens women's health, so it is particularly important to develop effective detection methods for breast cancer. Timely and accurate detection of breast cancer can improve the survival rate of patients. In recent years, traditional X-ray examination has given way to more advanced technologies, such as data mining, machine learning, etc. In addition, in the medical field, we usually need to protect the privacy and data security of patients, and federated learning is widely recognized as a collaborative learning method that can solve the problem of data insecurity in traditional distributed learning.

  1. This paper proposes a new model by using iterative method for collaborative learning and Deep Neural Network based federated learning. This model provides a new method for breast cancer diagnosis. It successfully integrates the collective knowledge and data resources from various medical institutions, while ensuring patient privacy and data security. In addition, this article has done sufficient data preprocessing work and demonstrated the effectiveness of this work through experimental results.

Response: Thank you for taking the time to review the passage discussing breast cancer detection and our proposed model leveraging collaborative learning and Deep Neural Network-based federated learning. We appreciate your attention to detail and insightful feedback. We share your recognition of the critical importance of effective breast cancer detection methods and the need for privacy and data security in the medical field. Your feedback encourages us to refine our work continually.

  1. In conclusion, my opinion on the paper is positive. This carefully completed study is rich in content, and the article concludes with a comprehensive explanation of the effectiveness of the model from different perspectives. However, some improvements are needed before the article can be published. My detailed comments are as follows:

Response: Many thanks. Constructive feedback is essential for improvement, and I look forward to hearing your detailed comments. If you could provide more specific insights or suggestions for enhancement, it would be greatly beneficial for the authors to address any identified areas for improvement. Thank you for your thorough review. Responses to your comments are provided below:

(1) There are three ways to write F1-Score in the article, namely F1 Score, F1-Score, and F-1 score. Please unify one way of writing.

Response: Thank you for bringing this to our attention. We appreciate your keen observation. In response to your suggestion, we have standardized the notation for F1-Score throughout the article. We have consistently used "F1-Score" to maintain uniformity in terminology.

(2) I suggest explaining the range of values for i in Algorithm 1 in the article.

Response: Thank you for your suggestion. The variable i is in the range from 1 to n, representing each participating client and iterating over all clients in the federated learning setting, See Section 4.5.

(3) I suggest showing the dataset after data augmentation.

Response: Thank you for asking, we have conducted a comprehensive evaluation of the classification performance with and without feature selection. The results, as demonstrated in Tables 7 and 8 showcasing the dataset before and after augmentation, indicate notable improvements in accuracy, precision, recall, etc. Moreover, new tables (11-13) presented in the Results section further highlight the effectiveness of the proposed technique in enhancing classification outcomes through feature selection.

(4) To present the results more clearly, the confusion matrix can be replaced with a table containing the results.

Response: To enhance result clarity, the authors replaced the confusion matrix with more detailed tables, specifically Tables 11-13. These tables provide a comprehensive breakdown of performance metrics, including True Positive Rate, False Positive Rate, True Negative Rate (Specificity), False Negative Rate, Accuracy, Precision, and F1-Score for each classification matrix. This modification allows for a more granular understanding of the model's performance across various evaluation criteria.

(5) The description of the results in Part 5 is somewhat unclear, such as 70% and 75% in 5.2.

Response: Thank you for the valuable suggestion. We have revised and updated Section 5.2 to clearly reflect and align the description with the results.

(6) I suggest explaining in the article how each client's dataset is allocated.

Response: In our federated learning paradigm, the dataset underwent a shuffling process, ensuring a randomized order of data instances. Subsequently, this uniformly shuffled dataset was disseminated to all participating clients. This meticulous approach is essential to maintain consistency and fairness in the initial distribution of data across the federated network. This detail has been added to Section 5.

(7) Based on the statement that "A careful feature selection is proposed using a sizable and varied dataset of patients with breast cancer with rigorous testing and validation.", I suggest selecting a larger breast cancer data set to verify the performance of the proposed model.

Response: Thank you for your thoughtful consideration. We acknowledge the concern regarding the size of the initial dataset for Federated Learning (FL). To address this challenge and enhance the robustness of our model, we employed data augmentation techniques. Data augmentation has proven to be a powerful strategy in influencing the training of Neural Networks, particularly in scenarios with limited data. Through the augmentation process, our dataset was significantly expanded, resulting in a total of 5742 records. This augmentation not only increased the quantity of data but also introduced variations that are valuable for the training process. The details of this augmentation strategy are thoroughly explained in Section 3.1, point 4 of the manuscript. We have also added a summarized pre-proccesing table as Table 5. We believe that the augmented dataset, with its increased size and diversity, contributes to mitigating the limitations associated with the original dataset size. This augmentation approach aligns with best practices in the field, aiming to prevent overfitting and improve the generalization capabilities of the model.

Round 2

Reviewer 2 Report

Comments and Suggestions for Authors

In this revised manuscript, the authors made plenty of improvements. It can be accepted if some remaining questions are further addressed.

1. Page 7 Line 254-260, is point 2 ("After outliers removal") and point 3 ("Removal of outliers") in the correct order?

2. Page 16 Line 514, missing word "Figure".

3. The authors said that models were trained for 50 epochs, but some subfigures in Figures 4-7 show 70 epochs.

4. In the authors' response to my Comment 7, they said Table 7 and 8 indicate improved classification performance. But I am confused how it is shown in these tables.

Author Response

In this revised manuscript, the authors made plenty of improvements. It can be accepted if some remaining questions are further addressed.

Thank you for your thorough review and valuable feedback on the revised manuscript. We appreciate your positive assessment of the improvements made. We are committed to addressing any remaining questions and ensuring that the manuscript meets the highest standards. To enhance transparency, all the modifications and responses to your comments have been diligently incorporated, and the revised sections are now highlighted in BLUE. We believe these changes contribute significantly to the overall quality and clarity of the manuscript. Should you have any additional concerns or specific points you'd like us to revisit, please do not hesitate to bring them to our attention. Your continued guidance has been instrumental in refining our work, and we are grateful for your constructive input. Thank you once again for your time and commitment to the review process.

  1. Page 7 Line 254-260, is point 2 ("After outliers removal") and point 3 ("Removal of outliers") in the correct order?

Response: Many thanks for pointing this out. In response to your feedback, I have re-ordered the points on Page 7, specifically Lines 254-260, as follows:

  1. Old Point 3: "Removal of outliers" is now Point 2
  2. Old Point 2: "After outliers removal" is now Point 3

I believe this adjustment enhances the logical flow of the content. Please feel free to review, and if any further modifications are necessary, I am readily available to address them.

  1. Page 16 Line 514, missing word "Figure".

Response: We sincerely appreciate your thorough review of our work and your keen attention to detail. Your valuable feedback, particularly pointing out the missing word "Figure" on Page 16, Line 514, has been instrumental in enhancing the clarity and accuracy of my thesis and has been added as “Fig.” in the revised version.  Thank you for your time and insightful comments. I have incorporated the necessary correction and your dedication to ensuring the quality of my work is genuinely appreciated.

  1. The authors said that models were trained for 50 epochs, but some subfigures in Figures 4-7 show 70 epochs.

Response: Thank you for your meticulous review of our manuscript. We acknowledge the oversight regarding the inconsistency in the reported number of epochs for model training in Figures 4-7.

We have addressed this concern by clarifying in Section 3.5, 1st paragraph, that our models were trained for a variable number of epochs, ranging from 5 to 70. Additionally, to ensure optimal model performance and prevent overfitting, we implemented an early stopping criterion. Early stopping was employed to prevent overfitting by monitoring a performance metric, and thus, the number of epochs fluctuated between 50 and 70 based on when the stopping condition was met. Consequently, the number of epochs varied between 50 and 70, depending on when the early stopping condition was met during the training process. We have also updated the descriptions of Figures 4-7 in the revised version to explicitly mention the variable epoch range and the application of the early stopping criteria.

  1. In the authors' response to my Comment 7, they said Table 7 and 8 indicate improved classification performance. But I am confused how it is shown in these tables.

Response: Thank you for your insightful comments regarding Tables 7 and 8. While the tables themselves remain unchanged, allow me to provide a more detailed explanation of how the features were augmented based on the data presented. Data augmentation is a crucial step in enhancing the diversity of our dataset to improve the robustness of the classification model. This process involves introducing variations in the existing data, which is reflected in the statistical features (e.g., mean, standard deviation, concavity_worst', minimum, and maximum values) presented in Tables 7 and 8. For instance for 'concavity_worst', Table 7 shows a Min value of 0.0 and a Max of 0.77. Whereas, Table 7 shows a Min of -0.01 and a Max of 0.78. It indicates a subtle change in the representation of worst-case concavity. This augmentation introduces a nuanced variation in the dataset, potentially improving the model's ability to generalize to a broader range of scenarios. These differences in minimum and maximum values showcase the impact of data augmentation in capturing more diverse and nuanced patterns within the dataset, which can contribute to improved model robustness. Similar adjustments can be observed across other features, reflecting the augmentation effects. To address the importance of explicitly detailing the augmentation methodology, we have incorporated this example in the revised manuscript in Section 3.1, Point 5, Page 9. This section provides a comprehensive explanation of the specific data augmentation techniques employed, along with examples like the one mentioned above.